# Prompt Certified Machine Unlearning with Randomized Gradient Smoothing and Quantization

**Zijie Zhang**[1]    **Yang Zhou**[1]*  **Xin Zhao**[1]    **Tianshi Che**[1]    **Lingjuan Lyu**[2]*

[1]Auburn University, [2]Sony AI

zzz0092@auburn.edu, yangzhou@auburn.edu, cmkk684735@gmail.com,
tzc0029@auburn.edu, lingjuan.lv@sony.com

## Abstract

The right to be forgotten calls for efficient machine unlearning techniques that make trained machine learning models forget a cohort of data. The combination of training and unlearning operations in traditional machine unlearning methods often leads to the expensive computational cost on large-scale data. This paper presents a P̲rompt C̲ertified M̲achine U̲nlearning algorithm, PCMU, which executes one-time operation of simultaneous training and unlearning in advance for a series of machine unlearning requests, without the knowledge of the removed/forgotten data. First, we establish a connection between randomized smoothing for certified robustness on classification and randomized smoothing for certified machine unlearning on gradient quantization. Second, we propose a prompt certified machine unlearning model based on randomized data smoothing and gradient quantization. We theoretically derive the certified radius $R$ regarding the data change before and after data removals and the certified budget of data removals about $R$. Last but not least, we present another practical framework of randomized gradient smoothing and quantization, due to the dilemma of producing high confidence certificates in the first framework. We theoretically demonstrate the certified radius $R'$ regarding the gradient change, the correlation between two types of certified radii, and the certified budget of data removals about $R'$.

## 1  Introduction

In order to to respect the privacy, machine unlearning techniques aim to enable data owners to proactively remove their data and eliminate its influence from already trained machine learning model upon requests [10, 43, 111, 42, 49, 39, 50, 134, 97]. A straightforward solution is to retrain new models from scratch on the remaining/remembered data, without the knowledge of the removed/forgotten data, as if the retrained model has never seen the forgotten data. However, the naive method is impractical since it often encounters expensive cost over complex models (e.g., DNN) on large data.

This has motivated the recent study of resolving the inefficiency issue of the naive machine unlearning. Existing techniques can be broadly classified into two categories: (1) Exact unlearning algorithms aim to learn an unlearning model with the same performance as the above naive ones retraining from scratch by completely excluding the forgotten data from the training data [107, 12, 62, 17, 78, 85, 8, 108, 42, 10, 4, 7, 124, 15, 14] and (2) Approximate unlearning methods try to bring the parameters of the trained model closer to the naive ones through the relaxation of exact unlearning requirements [4, 48, 45, 122, 79, 87, 136, 56, 94, 64, 42, 49, 94, 110, 43, 44, 49, 96, 46, 37, 79, 81, 83, 126, 133, 45, 15, 143, 82]. Certified removal is a certified-removal mechanism that applies a Newton step on the model parameters that largely remove the influence of the deleted data points [49]. GKT is a zero-shot machine unlearning algorithm that imposes the constraint that zero training data

---

*Corresponding authors

is available to the unlearning algorithms [20]. Two recent studies propose online machine unlearning methods for linear regression models [78] and linear support vector machine models [17], for further improving the efficiency of machine unlearning. The former adapts users' requests to delete their data before a specific time bar. The latter conducts only the task of variable support vector machine.

Despite achieving remarkable success, most of the above machine unlearning methods consist of two sequential operations: (1) Training: train a model on the complete training data and (2) Unlearning: generate an unlearning model from the former. The combination of two operations is computationally expensive when training complex models over large datasets. In addition, they often sequentially redo the unlearning operations one by one, when addressing a series of machine unlearning requests.

Randomized smoothing has achieved the state-of-the-art certified robustness guarantees against worst-case attacks by smoothing with isotropic Gaussian distribution [69, 23, 70, 76, 104, 72, 144, 58, 6, 38, 141, 92, 88, 33, 13, 142, 119, 1, 16, 53, 86]. Specially, it takes a base classifier $f(x)$ as an input, and outputs a smooth classifier $g(x) = \underset{c \in \mathcal{Y}}{\arg\max} \; \underset{\varepsilon \sim \mathcal{D}}{\mathbb{P}} (f(x + \varepsilon) = c)$ by averaging its prediction over isotropic Gaussian noise $\mathcal{D} = \mathcal{N}\left(0, \sigma^2 I\right)$ of the input data $x$ within its neighborhood $\varepsilon$. It provides a tight certified robustness guarantee: $g(x)$ can always return the most probable class $c_A$, i.e., $g(x + \delta) = c_A$ for all $\|\delta\|_2 < R$, as long as the perturbation $\delta$ is within the certified radius $R$.

This motivates us to establish a connection between randomized smoothing for certified robustness on classification and randomized smoothing for certified machine unlearning on gradient quantization. We analogize the data removals on the entire training data (i.e., the perturbations on the entire data) in the machine unlearning to the adversarial attacks (i.e., the perturbations on the data samples) in the certified robustness and liken the output quantized gradients in the former to the output discrete class labels in the latter. Since the output class labels in the latter through randomized smoothing are able to keep unchanged and correct against adversarial attacks within the certified radius, it is highly possible that the output quantized gradients in the former through randomized smoothing can keep unchanged against data removals within the certified budget, which implies that the learnt model shares the same gradients (and parameters) with the naive one retrained on only the remembered data.

Instead of performing two sequential operations of training and unlearning, this work directly trains an unlearning model in advance, without the knowledge of the forgotten data, based on randomized data smoothing and gradient quantization. We propose to execute the randomized smoothing on the average $\overline{x}$ of data samples. When there are data removal requests in the training data, $\overline{x}$ will be updated with a new average $\overline{x}'$, which can be treated as a perturbed version of $\overline{x}$. As the class labels lie in a (countable) small discrete space but the gradients lie in a large continuous space, we propose a gradient quantization technique to produce discrete gradients in a three-class space $\{-1, 0, 1\}$. The randomized smoothing method needs to sample a large number of points surrounding the input data for producing high confidence certificates, e.g., $10^5$ samples for $99.9\%$ confidence [23]. In our context, the outputs of randomized smoothing are high-dimensional gradients. We propose to utilize the Taylor expansion to approximate the output gradients of the sampled points for avoiding expensive gradient computation. We theoretically derive the error introduced by the Taylor expansion, the certified radius $R$ regarding the perturbation surrounding $\overline{x}$ and the certified budget $B$ of data removals (i.e., the maximally allowed amount of escaped data samples).

Notice that the response to the data removals is to erase the data samples and their associated labels together. It is necessary to smooth on both of them. However, for a given dataset, the relationship between the samples and labels is a multi-valued non-continuous mapping, which is a non-integrable function. This results in the dilemma of producing high confidence certificates, even with sampling and estimation techniques. Therefore, we propose a feasible solution based on randomized gradient smoothing and gradient quantization. We theoretically demonstrate the certified radius $R'$ regarding the gradient perturbations. Most importantly, we recognize the correlations between two types of radii $R$ and $R'$ and between $B$ and $R$, which are used to derive the certified budget $B'$ about $R'$. We further integrate the model training, randomized gradient smoothing, and gradient quantization into a unified framework for directly training a machine unlearning model with the data removal certificates as a guidance, for guaranteeing that the model parameters and gradients keep unchanged against the data removals within the certified budget.

In comparison with existing machine unlearning techniques, our randomized gradient smoothing and gradient quantization method exhibits three compelling advantages: (1) It simultaneously executes the training and unlearning operations, which is able to dramatically improve the unlearning efficiency

for complex models on large-scale data; (2) The one-time operation of simultaneous training and unlearning can provide the timely response to a series of machine unlearning requests, as long as the actual data removals are below the certified budget of data removals; and (3) It is agnostic to the removed/forgotten data before performing the unlearning operation.

Empirical evaluation on real datasets demonstrates the superior performance of our PCMU model against several state-of-the-art machine unlearning methods on image classification. More experiments, implementation details, and hyperparameter setting are presented in Appendices A.5-A.7.

## 2 Background

### 2.1 Randomized Smoothing for Certified Robustness

Randomized smoothing aims to build a smoothed classifier $g$ from a base classifier $f$ that maps inputs $x \in \mathbb{R}^d$ to classes $c \in C$.

$$g(x) = \operatorname*{argmax}_{c \in \mathcal{Y}} \mathbb{P}_{\varepsilon \sim \mathcal{D}} (f(x + \varepsilon) = c) \tag{1}$$

where $\mathcal{D} = \mathcal{N}\left(0, \sigma^2 I\right)$ is a Gaussian probability distribution in $\mathbb{R}^d$ for randomized smoothing. $g$ returns whichever class $f$ is most likely to return when $x$ is perturbed by noise $\varepsilon$.

Let $p_c(x)$ be the output probability of $f$ over class $c$, i.e., $p_c(x) = \mathbb{P}_{\varepsilon \sim \mathcal{D}} (f(x + \varepsilon) = c)$. Without loss of generality, we assume that $p_A(x)$ and $p_B(x)$ are the probabilities on the most probable class $c_A$ and the runner-up class $c_B$ respectively. If $\mathbb{P}\left(f(x + \varepsilon) = c_A\right) \geq \underline{p_A} \geq \overline{p_B} \geq \max_{c \neq c_A} \mathbb{P}(f(x + \varepsilon) = c)$, where $\underline{p_A}(x)$ is a lower bound of $p_A(x)$ and $\overline{p_B(x)}$ is an upper bound of $p_B(x)$, then $g(x + \delta) = c_A$ for $\forall \delta \in \mathbb{R}^d, \|\delta\|_2 \leq R$. In this case, the smoothed classifier $g$ can always output the correct prediction as long as the perturbation $\delta$ is within a certified $l_2$-norm radius of $R$.

**Theorem 1.** *Let $f : \mathbb{R}^d \to \mathcal{Y}$ be any deterministic or random function, and let $\varepsilon \sim \mathcal{N}\left(0, \sigma^2 I\right)$. Let $g$ be defined as in (1). Suppose $c_A \in \mathcal{Y}$ and $\underline{p_A}, \overline{p_B} \in [0, 1]$ satisfy [23]:*

$$\mathbb{P}\left(f(x + \varepsilon) = c_A\right) \geq \underline{p_A} \geq \overline{p_B} \geq \max_{c \neq c_A} \mathbb{P}(f(x + \varepsilon) = c) \tag{2}$$

*Then $g(x + \delta) = c_A$ for all $\|\delta\|_2 < R$, where*

$$R = \frac{\sigma}{2} \left(\Phi^{-1}\left(\underline{p_A}\right) - \Phi^{-1}\left(\overline{p_B}\right)\right) \tag{3}$$

where $\Phi^{-1}$ is the inverse of the standard Gaussian CDF.

### 2.2 Machine Unlearning

Machine unlearning aims to enable the trained models to forget what has been learned from the data to be removed. Specifically, given a training dataset of $N$ samples $D = \{x_i, y_i\}_{i=1}^N$. Each sample $x_i \in \mathbb{R}^d$ is associated with a label $y_i \in \mathcal{Y} = \{1, 2, ..., Y\}$, where $Y$ is the number of classes. A classification model $M(D)$ is trained on the complete training dataset $D$.

The users can submit a data removal request at any time. Thus, the complete training data $D$ is partitioned into two subsets: $D_f \subseteq D$ denoting the data which we wish the classification model to forget and $D_r \subseteq D$ specifying the data which we want the model to remember ($D = D_f \cup D_r$). The goal of machine unlearning is to unlearn the forgotten data $D_f$, i.e., eliminate the influence of $D_f$ from $M(D)$. A straightforward solution is to use the remembered data $D_r$ as the training data to retrain a new classification model $M_r(D_r)$ from scratch. However, this naive method is often time-consuming over large-scale datasets. An efficient algorithm is to directly generate a sanitized model $M_u(D, D_f, M)$ from the deployed model $M(D)$ that approximates $M_r(D_r)$, i.e.,

$$M_u(D, D_f, M) \approx M_r(D_r) \tag{4}$$

## 3  Randomized Data Smoothing and Gradient Quantization

The idea of this work is to establish a connection between randomized smoothing for certified robustness on classification and randomized smoothing for certified machine unlearning on gradient quantization. By leveraging the theory of randomized smoothing and gradient quantization, we theoretically derive the certified radius $R$ regarding the perturbation surrounding the data average $\bar{x}$ and the certified budget $B$ of data removals.

The gradient $G(x, y) \in \mathbb{R}^T$ of a machine learning model is given as follows.

$$G(x, y) = \frac{\partial \mathcal{L}(x, y; w)}{\partial w} \tag{5}$$

where $\mathcal{L}$ is the loss function, e.g., cross-entropy for image classification. $w$ is the model parameter.

We propose to quantize each dimension $t$ ($t = 1, \cdots, T$) of the continuous gradient $G(x, y) \in \mathbb{R}^T$ over a discrete three-class space $\{-1, 0, 1\}$, for mimicking the classification in the randomized smoothing for certified robustness.

$$Q(t) = Softmax([-|t - \sigma^2|, -|t|, -|t + \sigma^2|]) \tag{6}$$

where $Q(t)$ maps a gradient dimension $t$ to a three-dimensional vector $[-|t - \sigma^2|, -|t|, -|t + \sigma^2|]$, where each component denotes the similarity score between $t$ and $-\sigma^2$, 0, or $\sigma^2$. $\sigma$ is the standard deviation of the Gaussian probability distribution in the randomized smoothing and also serves as a quantization threshold in our method. The details of the selection of quantization threshold are presented in Appendix A.2. Therefore, all $T$ gradient dimensions are partitioned into three intervals: $(-\infty, -\sigma^2/2]$ that comes near to $-\sigma^2$, $[-\sigma^2/2, \sigma^2/2]$ that is closer to 0, and $[\sigma^2/2, \infty)$ that approaches $\sigma^2$. Each component in $Q(t)$ with the Softmax function also represents the probability of the gradient dimension $t$ belonging to classes -1, 0, or 1. The most probable class $c_A \in \{-1, 0, 1\}$ in $Q(t)$ is assigned to dimension $t$ as a final quantized gradient dimension.

We represent the composition $F(x, y)$ of gradient computation and quantization as follows.

$$F(x, y) = Q(G(x, y)) \tag{7}$$

We use $F^t(x, y)$ to denote the output three-dimensional quantization vector of the $t^{th}$ ($t = 1, \cdots, T$) dimension of the gradient $G(x, y)$ and use $F_c^t(x, y)$ to represent the $c^{th}$ ($c \in \{-1, 0, 1\}$) component of $F^t(x, y)$.

As the data removal is treated as the noise on the entire training data, we use the average $\bar{x}$ of all data samples to represent the entire training data.

$$\bar{x} = \frac{1}{N} \sum_{x_i \in D} x_i, \quad \bar{y} = \frac{1}{N} \sum_{y_i \in D} y_i \tag{8}$$

where $\bar{y}$ is the average of the class labels of all data samples.

The randomized data smoothing for certified unlearning on gradient quantization is defined below.

$$S^t(\bar{x}, \bar{y}) = \underset{c \in \{-1, 0, 1\}}{\operatorname{argmax}} \underset{\varepsilon_x, \varepsilon_y \sim \mathcal{D}}{\mathbb{P}} (F^t(\bar{x} + \varepsilon_x, \bar{y} + \varepsilon_y) = c) \tag{9}$$

where $\mathcal{D} = \mathcal{N}(0, \sigma^2 I)$ is a Gaussian distribution. $S^t$ is a smoothed version of a base gradient quantizatier $F^t$ that maps the $t^{th}$ gradient dimension about inputs $(\bar{x}, \bar{y})$ to gradient classes $c \in \{-1, 0, 1\}$. $S^t$ returns whichever class $F^t$ is most likely to return when $(\bar{x}, \bar{y})$ is perturbed by $(\varepsilon_x, \varepsilon_y)$.

The randomized smoothing method needs to sample a large number of points surrounding the input data for producing high confidence certificates, e.g., $10^5$ samples for 99.9% confidence [23]. In our context, computing the gradients for massive samples is extremely inefficient. We utilize the Taylor expansion to approximate the output gradients of sampled points, based on the outputs from

the original sample $(\bar{x}, \bar{y})$. For each quantization component $F_c^t(\hat{\bar{x}}, \hat{\bar{y}})$ for a sample in a Gaussian distribution surrounding $(\bar{x}, \bar{y})$, where $\hat{\bar{x}} = \bar{x} + \varepsilon_x, \hat{\bar{y}} = \bar{y} + \varepsilon_y, \varepsilon_x, \varepsilon_y \sim \mathcal{D}$, we take the Taylor expansion of $F_c^t(\hat{\bar{x}}, \hat{\bar{y}})$ at $(\bar{x}, \bar{y})$ as follows.

$$
\begin{aligned}
F_c^t(\hat{\bar{x}}, \hat{\bar{y}}) =& F_c^t(\bar{x}, \bar{y}) + \frac{\partial F_c^t}{\partial x}(\bar{x}, \bar{y})(\hat{\bar{x}} - \bar{x}) + \frac{\partial F_c^t}{\partial y}(\bar{x}, \bar{y})(\hat{\bar{y}} - \bar{y}) + \\
&+ \frac{1}{2}\{\frac{\partial^2 F_c^t(\bar{x}, \bar{y})}{\partial x^2}(\hat{\bar{x}} - \bar{x})^2 + 2\frac{\partial^2 F_c^t(\bar{x}, \bar{y})}{\partial x \partial y}(\hat{\bar{x}} - \bar{x})(\hat{\bar{y}} - \bar{y}) + \frac{\partial^2 F_c^t(\bar{x}, \bar{y})}{\partial y^2}(\hat{\bar{y}} - \bar{y})^2\} \\
&+ \cdots + O_j(\hat{\bar{x}}, \hat{\bar{y}})
\end{aligned}
\tag{10}
$$

where $O_j(\hat{\bar{x}}, \hat{\bar{y}}) = \frac{1}{(j+1)!}\{\sum \cdots \sum \frac{\partial^k F_c^t(\xi)}{\partial x^k \partial y^{j+1-k}}(\hat{\bar{x}} - \bar{x})^k(\hat{\bar{y}} - \bar{y})^{j+1-k}\}, \xi \in ((\hat{\bar{x}}, \bar{x}), (\hat{\bar{y}}, \bar{y})), j = 3, \cdots, \infty$, and $k = 1, 2, 3$.

The following theorem derives the error introduced by the Taylor expansion.

**Theorem 2.** *The error introduced by the Taylor expansion of $F_c^t(\hat{\bar{x}}, \hat{\bar{y}})$ at $(\bar{x}, \bar{y})$ is*

$$
\epsilon \leq \sum_{j=0}^{\infty} \|\frac{L_{j+1}}{(j+1)!}\| \cdot \|\sigma M\|^{j+1}
\tag{11}
$$

*where $L_j = \max_{k=1,\cdots,j} \frac{\partial^j h_i}{\partial x^k \partial y^{j-k}}$ and $M$ is the number of sampled points.*

*Please refer to Appendix A.4 for detailed proof of Theorem 2.*

Notice that

$$
\mathbb{P}(F^t(\hat{\bar{x}}, \hat{\bar{y}}) = c) = \int_{\hat{\bar{x}}} \int_{\hat{\bar{y}}} \mathbb{P}(F_c^t(\hat{\bar{x}}, \hat{\bar{y}})) d\hat{\bar{x}} d\hat{\bar{y}} \geq \max_{c \in \{-1,0,1\}} F_c^t(\hat{\bar{x}}, \hat{\bar{y}})
\tag{12}
$$

Thus, we derive the probabilities $\underline{p_A}$ and $\overline{p_B}$ on the most probable class $c_A$ and the runner-up class $c_B$ in the randomized smoothing for certified machine unlearning ($c_A, c_B \in \{-1, 0, 1\}$), based on the error introduced by the Taylor expansion.

$$
\underline{p_A} = \max_{c \in \{-1,0,1\}} \mathbb{P}(F_c^t(\hat{\bar{x}}, \hat{\bar{y}}) - \epsilon) = \int_{\hat{\bar{x}}} \int_{\hat{\bar{y}}} \mathbb{P}(F_c^t(\hat{\bar{x}}, \hat{\bar{y}}) - \epsilon) d\hat{\bar{x}} d\hat{\bar{y}}, \ c = c_A
\tag{13}
$$

$$
\overline{p_B} = \max_{c \neq c_A} \mathbb{P}(F_c^t(\hat{\bar{x}}, \hat{\bar{y}}) + \epsilon) = \int_{\hat{\bar{x}}} \int_{\hat{\bar{y}}} \mathbb{P}(F_c^t(\hat{\bar{x}}, \hat{\bar{y}}) + \epsilon) d\hat{\bar{x}} d\hat{\bar{y}}, \ c \neq c_A
\tag{14}
$$

Notice that the correlation between the data $\bar{x}$ and its classes $\bar{y}$ is fixed for a given dataset. We denote this correlation as $\bar{y} = H(\bar{x})$ where $H : \mathbb{R}^d \mapsto C$. Thus, the randomized smoothing for certified machine unlearning in Eq.(9) is rewritten as an equivalent one as follows.

$$
\tilde{S}^t(\bar{x}) = \underset{c \in \{-1,0,1\}}{\operatorname{argmax}} \underset{\varepsilon \sim \mathcal{D}}{\mathbb{P}}(\tilde{F}^t(\bar{x} + \varepsilon) = c)
\tag{15}
$$

where $\tilde{F}^t(\bar{x}) = F^t(\bar{x}, H(\bar{x})) = F^t(\bar{x}, \bar{y})$.

Based on the computed $\underline{p_A}$, $\overline{p_B}$, and $F'(\bar{x})$, we can obtain the certified radius $R$ regarding the perturbation surrounding $\bar{\bar{x}}$ and the certified budget $B$ of data removal.

**Theorem 3.** *Let $\varepsilon \sim \mathcal{D} = \mathcal{N}\left(0, \sigma^2 I\right)$ and $\tilde{S}^t(\bar{x}) = \underset{c \in \{-1,0,1\}}{\operatorname{argmax}} \underset{\varepsilon \sim \mathcal{D}}{\mathbb{P}}(\tilde{F}^t(\bar{x} + \varepsilon) = c)$. Suppose that for a specific $\bar{x} \in \mathbb{R}^d$, there exist $c_A \in \{-1, 0, 1\}$ and $\underline{p_A}, \overline{p_B} \in [0, 1]$ such that:*

$$
\mathbb{P}\left(\tilde{F}^t(\bar{x} + \varepsilon) = c_A\right) \geq \underline{p_A} \geq \overline{p_B} \geq \max_{c \neq c_A} \mathbb{P}(\tilde{F}^t(\bar{x} + \varepsilon) = c)
\tag{16}
$$

*Then $\tilde{S}^t(\bar{x} + \delta) = c_A$ for all $\|\delta\|_2 < R$, where*

$$R = \frac{\sigma}{2}\left(\Phi^{-1}\left(\underline{p_A}\right) - \Phi^{-1}\left(\overline{p_B}\right)\right) \tag{17}$$

*where $\Phi^{-1}$ is the inverse of the standard Gaussian CDF.*

**Theorem 4.** *Let $R$ be the certified radius of $\bar{x} \in \mathbb{R}^d$ based on $\tilde{S}^t(\bar{x}) = \underset{c\in\{-1,0,1\}}{\operatorname{argmax}} \underset{\varepsilon\sim\mathcal{D}}{\mathbb{P}}(\tilde{F}^t(\bar{x}+\varepsilon) = c)$, then the certified budget of data removals is*

$$B \leq N - \frac{9d\sigma^2}{R^2} \tag{18}$$

*Please refer to Appendix A.4 for detailed proof of Theorems 3 and 4.*

The above method is effective for certified machine unlearning, but it is computationally expensive to calculate high-dimensional double integrals in $\underline{p_A}$ and $\overline{p_B}$. We can reduce the double integrals to the single integrals through $\bar{y} = H(\bar{x})$. However, $H$ is essentially a multi-valued non-continuous mapping, which is not an integrable function and makes the above method impractical.

$$\underline{p_A} = \int_{\hat{\bar{x}}}\int_{\hat{\bar{y}}} \mathbb{P}(F_c^t(\hat{\bar{x}}, \hat{\bar{y}}) - \epsilon)d\hat{\bar{x}}d\hat{\bar{y}} = \int_{\hat{\bar{x}}} \mathbb{P}(F_c^t(\hat{\bar{x}}, H(\hat{\bar{x}})) - \epsilon)d\hat{\bar{x}}, \ c = c_A \tag{19}$$

## 4 Randomized Gradient Smoothing and Quantization

In order to avoid the dilemma of computing practical $\underline{p_A}$, $\overline{p_B}$ and $R$, we propose a feasible solution based on randomized gradient smoothing and gradient quantization. We theoretically demonstrate the certified radius $R'$ regarding the gradient perturbations. We recognize the correlations between $R$ and $R'$ and between $B$ and $R$, which are used to derive the certified budget $B'$ about $R'$.

We first calculate the gradient $G(x_i, y_i)$ in terms of each sample $(x_i, y_i)$ and the gradient average $\bar{G}$.

$$\bar{G} = \frac{1}{N} \sum_{(x_i,y_i)\in D} G(x_i, y_i) \tag{20}$$

The randomized gradient smoothing for certified unlearning on gradient quantization is defined below.

$$S^{t\prime}(\bar{G}) = \underset{c\in\{-1,0,1\}}{\operatorname{argmax}} \underset{\varepsilon\sim\mathcal{D}}{\mathbb{P}}(Q^t(\bar{G}+\varepsilon) = c) \tag{21}$$

where $\mathcal{D} = \mathcal{N}\left(0, \sigma^2 I\right)$ is a Gaussian distribution. $S^{t\prime}$ is a smoothed version of a base gradient quantizatier $Q^t$ that maps each dimension $t$ of the gradient $\bar{G}$ to gradient classes $c \in \{-1,0,1\}$. $S^{t\prime}$ returns whichever gradient class $Q^t$ is most likely to return when $\bar{G}$ is perturbed by noise $\varepsilon$.

We compute the probabilities over three intervals of $(-\infty, -\sigma^2/2]$, $[-\sigma^2/2, \sigma^2/2]$, and $[\sigma^2/2, \infty)$.

$$P = \left\{ \int_{\frac{\sigma^2}{2}-\hat{G}^t}^{\infty} \frac{1}{\sigma\sqrt{2\pi}}e^{-\frac{z^2}{2\sigma^2}}\,dz, \int_{-\frac{\sigma^2}{2}-\hat{G}^t}^{\frac{\sigma^2}{2}-\hat{G}^t} \frac{1}{\sigma\sqrt{2\pi}}e^{-\frac{z^2}{2\sigma^2}}\,dz, \int_{-\infty}^{-\frac{\sigma^2}{2}-\hat{G}^t} \frac{1}{\sigma\sqrt{2\pi}}e^{-\frac{z^2}{2\sigma^2}}\,dz \right\} \tag{22}$$

Now, we can directly generate the corresponding probabilities $\underline{p'_A}$ and $\overline{p'_B}$.

$$\underline{p'_A} = \max P, \ \overline{p'_B} = \max\left\{P - \{\underline{p'_A}\}\right\} \tag{23}$$

By following the similar strategy in Theorem 3, we can derive the corresponding certified radius $R'$. Namely, $S^{t\prime}(\bar{G} + \delta) = c_A$ for all $\|\delta\|_2 < R'$, where

$$R' = \frac{\sigma}{2}\left(\Phi^{-1}\left(\underline{p'_A}\right) - \Phi^{-1}\left(\overline{p'_B}\right)\right) \tag{24}$$

However, it is difficult to obtain the corresponding certified budget of data removal from $R'$, since $R'$ is related to the perturbations over the gradient $\bar{G}$, instead of the data $\bar{x}$. The following theorem demonstrates the correlation between two types of radii $R$ and $R'$.

**Theorem 5.** *Let $R$ and $R'$ be the certified radii of the above two algorithms respectively and $L$ be the Lipschitz constant of gradient $G(x, y) \in \mathbb{R}^T$, then*

$$R \geq \frac{\sqrt{T}}{L}R' \tag{25}$$

*By combining Theorems 4 and 5 together, we derive the certified budget $B'$ of data removal from $R'$.*

$$B' \leq N - \frac{36dL^2}{T(\Phi^{-1}(\underline{p_{A'}}) - \Phi^{-1}(\overline{p_{B}'}))^2} \tag{26}$$

In addition, we conduct the convergence analysis of our prompt certified machine unlearning algorithm based on randomized gradient smoothing and quantization.

**Theorem 6.** *Let $S^{t\prime}(\bar{G})$ be the randomized gradient smoothing for certified machine unlearning on gradient quantization, $L$, $L_1$, and $L_2$ be the Lipschitz constants of $G$, $Q^t$, and $S^{t\prime}$ respectively, i.e.,*

$$||\nabla S^{t\prime}(a) - \nabla S^{t\prime}(b)||_2 \leq L_2 L_1 L||a - b||_2 \; for \; any \; a, b \tag{27}$$

*If we run gradient descent for $k$ iterations with a fixed step size $s \leq \frac{1}{L_2 L_1 L}$, it will yield a solution $S^{t\prime(k)}$ which satisfies*

$$S^{t\prime}(q^{(k)}) - S^{t\prime}(q^*) \leq \frac{||q^{(0)} - q^*||_2^2}{2sk} \tag{28}$$

*where $S^{t\prime}(q^{(0)})$ is the initial solution and $S^{t\prime}(q^*)$ is the local optimal solution.*

*This means that gradient descent is guaranteed to converge and that it converges with rate $\mathcal{O}(1/k)$.*

*Please refer to Appendix A.4 for detailed proof of Theorems 5 and 6.*

Finally, we integrate the model training for a specific learning task (e.g., image classification) randomized gradient smoothing, and gradient quantization into a unified framework for directly training a machine unlearning model with the data removal certificates as a guidance, for guaranteeing that the model parameters and gradients keep unchanged against the data removals within the certified budget. The corresponding parameter update is given below.

$$w = w - \eta[S^{1\prime}(\bar{G}), \cdots, S^{T\prime}(\bar{G})] \tag{29}$$

where $w$ is the model parameter and $\eta$ is a learning rate.

## 5 Experimental Evaluation

In this section, we have evaluated the effectiveness of our PCMU model and other comparison methods for machine unlearning over three popular image classification datasets: Fashion-MNIST [138, 50, 37], CIFAR-10 [66, 43, 44, 121, 50, 37], and SVHN [95, 49, 7]. We train the classifiers on the training set and test them on the test set for three datasets. We train a convolutional neural network (CNN) on Fashion-MNIST for clothing classification. We train LeNet over CIFAR-10 for image classification. We apply the ResNet-18 architecture on SVHN for street view house number identification. We

Table 1: Performance with 10% data removal and CNN on Fashion-MNIST

| Metric | Performance | | | | Runtime (s) | | |
|---|---|---|---|---|---|---|---|
| | $Accuracy$ | $Error_t$ | $Error_r$ | $Error_f$ | Training | Unlearning | Total |
| Retrain | 88.50 | 11.50 | 8.92 | 11.41 | 687 | 629 | 1,316 |
| Fisher | 86.23 | 13.77 | 12.61 | 13.33 | 671 | 2,015 | 2,686 |
| certified removal | 77.70 | 22.30 | 90.11 | 89.87 | 719 | 181 | 900 |
| DeltaGrad | 84.22 | 15.78 | 14.36 | 15.27 | 553 | 141 | 694 |
| NTK | 86.02 | 13.98 | 12.81 | 13.25 | 671 | 1,879 | 2,550 |
| Unrolling SGD | 83.44 | 16.56 | 41.13 | 41.00 | **356** | 63 | **419** |
| SISA | 84.46 | 15.54 | 14.59 | 14.52 | 1,419 | 1,387 | 2,806 |
| Adaptive Unlearning | 86.02 | 13.98 | 12.80 | 13.25 | 1,537 | 1,481 | 3,018 |
| FedEraser | 69.86 | 30.14 | 29.88 | 28.14 | 677 | 608 | 1,285 |
| MCMC unlearning | 85.29 | 14.71 | 6.62 | 58.73 | 621 | 803 | 1,424 |
| PCMU | **88.34** | **11.66** | **10.68** | **10.71** | 802 | **0** | 802 |

Table 2: Performance with 20% data removal and CNN on Fashion-MNIST

| Metric | Performance | | | | Runtime (s) | | |
|---|---|---|---|---|---|---|---|
| | $Accuracy$ | $Error_t$ | $Error_r$ | $Error_f$ | Training | Unlearning | Total |
| Retrain | 88.21 | 11.79 | 9.75 | 11.76 | 687 | 561 | 1,248 |
| Fisher | 86.02 | 13.98 | 12.80 | 13.25 | 827 | 1,939 | 2,766 |
| certified removal | 76.69 | 23.31 | 90.29 | 89.82 | 711 | 347 | 1,058 |
| DeltaGrad | 84.11 | 15.89 | 14.73 | 13.21 | 570 | 140 | 710 |
| NTK | 86.03 | 13.97 | 12.81 | 13.25 | 827 | 1,807 | 2,634 |
| Unrolling SGD | 85.56 | 14.44 | 38.27 | 37.50 | **371** | 65 | **436** |
| SISA | 83.90 | 16.10 | 14.64 | 14.71 | 1,419 | 1,351 | 2,770 |
| Adaptive Unlearning | 75.13 | 24.87 | 25.37 | 25.19 | 1,537 | 1,419 | 2,956 |
| FedEraser | 71.93 | 28.07 | 27.07 | 27.13 | 654 | 586 | 1,240 |
| MCMC unlearning | 84.52 | 15.48 | 6.95 | 63.02 | 995 | 778 | 1,773 |
| PCMU | **88.34** | **11.66** | **10.25** | **11.47** | 802 | **0** | 802 |

evaluate the performance of various machine unlearning methods on three datasets with the ratio of data removal between 5% and 20%. In this work, by following several representative machine unlearning methods [7, 50, 134], where each learning request is modeled as a random draw from the training data in terms of a uniform distribution. Given a ratio of data removal, the forgotten data $D_f$ are sampled uniformly from the complete training data $D$ with this ratio. The remaining dataset $D_r$ (i.e., $D = D_f \cup D_r$) will be considered as the remembered data. This sampling approach is more realistic since a removal request may be applied to any data examples with the same probability.

**Baselines.** We compare the PCMU model with nine state-of-the-art machine unlearning models. **Fisher** is a scrubbing procedure that removes information from the trained weights, without the need to access the original training data, nor to retrain the entire network [43]. **certified removal** provides a strong theoretical guarantee that a model from which data is removed cannot be distinguished from a model that never observed the data to begin with [49]. **DeltaGrad** is a rapid retraining machine learning model based on information cached during the training phase [136]. **NTK** removes dependency on a cohort of training data from a trained deep network that improves upon and generalizes previous methods to different readout functions [44]. **Unrolling SGD** is a taxonomy of approximate unlearning which concludes with verification error as a metric to study as it subsumes a large class of unlearning criteria [121]. **SISA** is a practical approach for unlearning that relies on data sharding and slicing to reduce the computational overhead of unlearning [7]. **Adaptive Unlearning** gives a general reduction from deletion guarantees against adaptive sequences to deletion guarantees against non-adaptive sequences. [50]. **FedEraser** is a federated unlearning methodology that can eliminate the influences of a federated client's data on the global model while significantly reducing the time consumption [79]. **MCMC unlearning** designs an MCMC influence function to characterize the knowledge learned from data, which then delivers the MCMC unlearning algorithm [37]. To our best knowledge, this work is the first to execute one-time operation of simultaneous training and unlearning in advance for a series of machine unlearning requests.

**Variants of PCMU model.** We evaluate two versions of PCMU to show the strengths of different techniques. PCMU-N uses the basic model with only the gradient quantization. PCMU operates with the full support of both the randomized gradient smoothing and the gradient quantization techniques. Notice that the gradient quantization is a necessary operation to convert continuous gradients to discrete gradient classes for the randomized gradient smoothing in our PCMU model. Thus, we cannot validate the version with only the randomized gradient smoothing.

**Evaluation metrics.** By following the same settings in several representative machine unlearning models [43, 44, 121, 37], we use four popular measures in machine unlearning to verify the

Table 3: Performance with 10% data removal and LeNet on CIFAR-10

| Metric | Performance | | | | Runtime (s) | | |
|---|---|---|---|---|---|---|---|
| | $Accuracy$ | $Error_t$ | $Error_r$ | $Error_f$ | Training | Unlearning | Total |
| Retrain | 64.31 | 35.69 | 27.44 | 36.16 | 846 | 745 | 1,591 |
| Fisher | 62.39 | 37.61 | 33.76 | 33.78 | 693 | 2189 | 2,882 |
| certified removal | 36.91 | 63.09 | 90.02 | 89.66 | 749 | 174 | 923 |
| DeltaGrad | 61.46 | 38.54 | 23.12 | 23.92 | 859 | 493 | 1,352 |
| NTK | 62.36 | 37.64 | 33.76 | 35.44 | 693 | 2,047 | 2,740 |
| Unrolling SGD | 59.69 | 40.31 | 41.69 | 49.80 | **511** | 168 | **679** |
| SISA | 58.01 | 41.99 | 34.01 | 34.40 | 1,594 | 1,533 | 3,127 |
| Adaptive Unlearning | 43.35 | 56.65 | 55.80 | 55.21 | 1,176 | 293 | 1,469 |
| FedEraser | 51.63 | 48.37 | 43.57 | 48.62 | 1,190 | 984 | 2,174 |
| MCMC unlearning | 60.70 | 39.30 | 4.53 | 26.98 | 1,322 | 718 | 2,040 |
| PCMU | **64.33** | **35.67** | **24.21** | **37.68** | 903 | **0** | 903 |

Table 4: Performance with 20% data removal and LeNet on CIFAR-10

| Metric | Performance | | | | Runtime (s) | | |
|---|---|---|---|---|---|---|---|
| | $Accuracy$ | $Error_t$ | $Error_r$ | $Error_f$ | Training | Unlearning | Total |
| Retrain | 63.29 | 36.71 | 24.59 | 36.89 | 846 | 673 | 1,519 |
| Fisher | Failed due to out of memory | | | | Failed due to out of memory | | |
| certified removal | 36.03 | 63.97 | 90.14 | 89.84 | 749 | 430 | 1,179 |
| DeltaGrad | 61.55 | 38.45 | 22.61 | 22.20 | 864 | 499 | 1,363 |
| NTK | Failed due to out of memory | | | | Failed due to out of memory | | |
| Unrolling SGD | 60.39 | 39.61 | 40.17 | 47.00 | **511** | 327 | **838** |
| SISA | 57.17 | 42.83 | 35.75 | 35.74 | 1,594 | 1,465 | 3,059 |
| Adaptive Unlearning | 41.23 | 58.77 | 57.52 | 59.22 | 1,176 | 321 | 1,497 |
| FedEraser | 51.66 | 48.34 | 48.53 | 50.14 | 1,190 | 980 | 2,170 |
| MCMC unlearning | 61.33 | 38.67 | 5.29 | 30.42 | 1,322 | 714 | 2,766 |
| PCMU | **64.33** | **35.67** | **25.18** | **35.32** | 903 | **0** | 903 |

performance of different methods: $Accuracy$, $Error_f$ (classification errors on the forgotten data $D_f$), $Error_r$ (errors on the remembered data $D_r$), and $Error_t$ (errors on the test data). Since the model $M_r(D_r)$ (**Retrain**) that uses only the remembered data $D_r$ as the training data retrained from scratch has never seen the forgotten data $D_f$, it is usually used as the gold standard for evaluating the performance of machine unlearning algorithms [43, 37]. Ideally, the accuracy and three errors of the unlearning models should match that of the retrained model $M_r(D_r)$.

**Machine unlearning accuracy with varying ratios of data removal.** Tables 1-4 exhibit the accuracy obtained by eleven machine unlearning approaches by varying the ratio of unlearning request / data removal between 10% and 20%. Retrain represents the model retrained on only the remembered data $D_r$ from scratch, without the knowledge of the forgotten data $D_f$. A machine unlearning algorithm with more similar performance to the Retrain model achieves a better unlearning result. It is observed that among ten approaches except the Retrain model, no matter how large the ratios of data removal are, the PCMU method achieves the closest accuracy to the Retrain model in all tests, showing the effectiveness of PCMU to the machine unlearning. Compared to the absolute performance difference between other baselines and the Retrain model, PCMU, on average, achieves at least 5.56% and 15.17% improvement of absolute accuracy difference on Fashion-MNIST and CIFAR-10 respectively. Notice that the accuracy and error on test data by our PCMU keep unchanged since it performs one-time operation of simultaneous training and unlearning for addressing multiple unlearning requests. In addition, the promising performance of PCMU over Fashion-MNIST and CIFAR-10 implies that PCMU has great potential as a general machine unlearning solution to other image datasets, which is desirable in practice.

**Machine unlearning error with varying ratios of data removal.** Tables 1-4 also show the classification errors on the deleted data $D_f$ ($Error_f$), errors on the remembered data $D_r$ ($Error_r$), and errors on the test data ($Error_t$) by eleven machine unlearning methods respectively. We have observed that the performance of our PCMU method behaves similarly and achieves at least 13.73% and 16.01% boost of absolute error difference on two datasets respectively. PCMU substantially outperforms the performance of other baselines in most experiments, especially on the CIFAR-10 dataset. In addition, the errors by our PCMU are not sensitive to the ratio of data removals. This is because that our PCMU method performs one-time operation of simultaneous training and unlearning when addressing a series of machine unlearning requests, as long as the ratio of actual data removals is below the certified budget of data removals in our PCMU. However, other baselines need to sequentially handle these machine unlearning requests one by one.

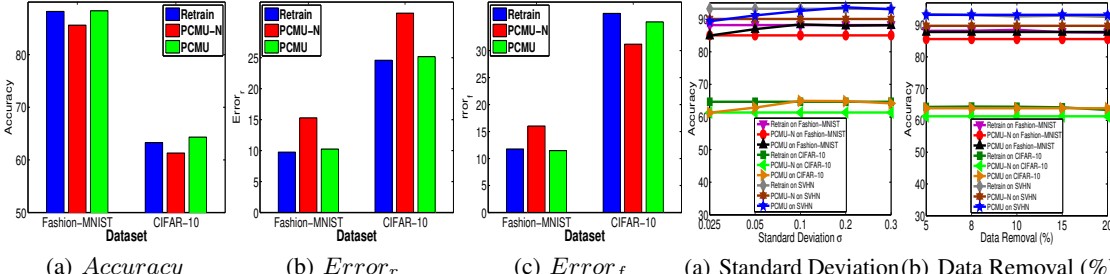

| (a) *Accuracy* | (b) $Error_r$ | (c) $Error_f$ | (a) Standard Deviation | (b) Data Removal (%) |

Figure 1: Performance of PCMU variants with 20% data removal on two datasets

Figure 2: Performance with varying parameters

**Ablation study.** Figure 1 exhibits the unlearning performance with the Retrain model and two variants of the PCMU model over two datasets of Fashion-MNIST and CIFAR-10 respectively. We have observed the exact PCMU achieves the closest accuracy and errors to the Retrain model over two datasets, which are obviously better than PCMU-N. A reasonable explanation is that our PCMU method utilizes the randomized gradient smoothing and gradient quantization techniques for supporting certified machine unlearning. It further uses the certificates as a guidance to train the machine unlearning model, for guaranteeing that the model parameters and gradients keep unchanged against the data removals within the certified budget.

**Running time.** Tables 1-4 report the running time achieved by all comparison methods over two dataset to produce machine unlearning results respectively. We observe that PCMU scales well with deep neural network architectures over different image datasets and shows good efficiency for machine unlearning. Our PCMU method achieves better efficiency than most baseline methods, except DeltaGrad and Unrolling SGD. As discussed above, our PCMU method performs one-time operation of simultaneous training and unlearning when addressing a series of machine unlearning requests. However, DeltaGrad and Unrolling SGD need to sequentially handle these machine unlearning requests one by one. This is clearly a computationally expensive process when the number of machine unlearning requests is huge.

**Impact of standard deviation.** Figure 2 (a) measures the performance effect of standard deviation of the Gaussian distribution in the randomized smoothing for machine unlearning by varying $\sigma$ from 0.025 to 0.3. Notice that the Retrain and PCMU-N models do not contain the module of randomized smoothing. Thus, their accuracy scores keep unchanged with varying $\sigma$. We have witnessed the performance curves by PCMU initially increase quickly and then become stable or even slight drop when $\sigma$ continuously increases. Initially, a large $\sigma$ can help utilize the strength of randomized gradient smoothing and quantization for directly training a machine unlearning model in advance. Later on, when $\sigma$ continues to increase and goes beyond some thresholds, the performance curves become stable. A rational guess is that after the data removals have been already certified at a certain threshold and considered in the training of machine unlearning models, our PCMU model is able to generate a good machine unlearning result. When $\sigma$ continuously increases, this does not affect the performance of machine unlearning any more.

**Impact of data removal ratio.** Figure 2 (b) evaluates the accuracy impact of data removal ratios varying from 5% to 20% on three datasets of Fashion-MNIST, CIFAR-10, and SVHN. It is observed that when changing data removal ratios, the accuracy by our PCMU model matchs well with that of the retrained model from scratch. The performance by our PCMU model keeps relatively stable, since our method directly trains a unlearning model based on the certified budget of data removals in advance and performs one-time operation of simultaneous training and unlearning, as long as the ratio of actual data removals is below the certified budget of data removals.

## 6   Conclusions

In this work, we have proposed a prompt certified machine unlearning algorithm that executes one-time operation of simultaneous training and unlearning in advance. First, we establish a connection between randomized smoothing for certified robustness on classification and randomized smoothing for certified machine unlearning on gradient quantization. Second, we propose a certified machine unlearning model based on randomized data smoothing and gradient quantization. Finally, we present another practical framework of randomized gradient smoothing and quantization, due to the dilemma of producing high confidence certificates in the first framework.

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
