# A Supplementary Materials

## A.1 Related Work

**Randomized Smoothing for Certified Robustness.** Trustworthy machine learning has attracted active research in recent years [99, 158, 150, 135, 159, 151, 101, 151, 149, 146, 80, 160, 61, 60, 117, 153, 118, 3, 155, 152, 157, 156, 154]. Certificate robustness techniques provide guarantees that an object with certifiable robustness is robust against any admissible attacks [130, 41, 19, 139, 57, 106, 103, 59, 93, 109, 73, 89, 36]. Existing certificate robustness approaches include satisfiability modulo theories-based methods [63, 55, 29, 11] and mixed integer-linear programming-based models [18, 84, 25, 34, 9]. The common characteristics of these methods is that the ever-increasing complexity of deep neural networks has made it difficult to scale these methods meaningfully to high-dimensional datasets like ImageNet. In addition, they are often applicable to certain specific neural networks. Convex relaxation-based algorithms [131, 132, 100, 105, 114, 115, 19, 116, 148], linear relaxation-based models [129, 127], abstract interpretation-based methods [40, 90, 113], interval-bound propagation-based algorithms [26, 47, 54], and Lipschitz constant-constrained approaches [21, 129, 27, 147, 128, 123, 24, 2, 22, 30, 77, 65] rely on over approximation loose too much precision and provide loose guarantees on the worst-case margin, even for networks trained to be amenable to certification. The robust training built upon the loose guarantees limits the improvement of model robustness.

In order to deal with the scalability and loose guarantee issues in certified robustness, a line of work has been introduced based on randomized smoothing [69, 23, 70, 76, 104, 72, 144, 58, 6, 38, 141, 92, 88, 33, 13, 142, 119, 1, 16, 53, 86]. Randomized smoothing-based methods have demonstrated exceptional scalability and applicability when defending against adversarial examples by relaxing exact certificates to high-confidence probabilistic ones [31, 58, 28, 51, 102, 72, 91, 32, 145]. We have witnessed many promising randomized smoothing-based methods for certifying $l_0$ [70, 72], $l_1$ [69, 76, 120], and $l_2$ robustness [75, 23, 104, 144, 74]. Several recent research efforts have studied and identified the inapplicability of randomized smoothing to high-dimensional problems for $l_p$ robustness against high-norm attacks when $p > 2$, especially $l_\infty$ robustness [68, 140, 67, 5, 137].

**Machine Unlearning.** Machine unlearning, also known as selective forgetting [10, 43, 111] or data removal/deletion [42, 49] in machine learning, aims to eliminate the effect of a subset of training data on the already trained model [39, 50, 134, 97]. Existing techniques on machine unlearning can be broadly classified into two categories below.

(1) Exact machine unlearning algorithms aim to learn an unlearning model with the same performance as the ones obtained with retraining from scratch by completely excluding the forgotten data from the training data. Many of existing methods mainly focus on producing the exact unlearning for simple models or under some specific conditions [107], like leave-one-out cross-validation for SVMs (Support Vector Machines) [12, 62, 17], efficient machine unlearning for linear models [78, 85], data removal-enabled random forests [8, 108], provably efficient data deletion in $K$-means clustering [42], and fast data deletion for Naïve Bayes based on statistical query learning which assumes the training data is in a decided order [10]. These unlearning models often fail to work on large-scale datasets with complex models. Several research efforts have tried to improve the efficiency of machine unlearning. Linear filtration is a novel algorithm for the sanitization of deep classification models that predict logits, after class-wide deletion requests [4]. SISA is a practical approach for unlearning that relies on data sharding and slicing to reduce the computational overhead of unlearning [7]. It is designed to achieve the largest improvements for stateful algorithms like stochastic gradient descent for deep neural networks. Ullah et al. proposed an efficient unlearning algorithm based on constructing a maximal coupling of Markov chains for the noisy SGD procedure [124]. GraphEraser is a machine unlearning method tailored to graph data with two novel graph partition algorithms and a learning-based aggregation method [15]. RecEraser is a general and efficient machine unlearning framework tailored to recommendation tasks with new data partition methods and an adaptive aggregation method to improve the global model utility. These designs make our RecEraser more suitable for recommendation tasks [14].

(2) Approximate machine unlearning methods try to bring the parameters of the trained model closer to naive ones retrained from scratch through the relaxation of the exact unlearning definitions and requirements [4, 48, 45, 122, 79, 87]. Most of these approaches make use of or reconstruct historical gradients and model weights to quickly eliminate the influence of samples that are requested to be deleted, such that the unlearned model cannot be distinguished from the model that was never

trained on the removed data [136, 56, 94, 64]. O-$k$-means and DC-$k$-means first introduced the definition of approximate unlearning based on the distance (or divergence) between distributions of the retrained model and the unlearnt one [42]. Subsequent works follow similar approximate definitions to provide certified unlearning guarantees for strongly-convex learning problems [49, 94, 110]. Several approaches proposed different Newton's methods to approximate retraining for convex models, e.g., linear regression, logistic regression, and the last fully connected layer of a neural network [43, 44, 49]. Some recent works have also studied machine unlearning in optimization-based Bayesian inference [96, 46] and sampling-based Bayesian inference [37]. Recent methods designed effective machine unlearning strategies in the setting of federated learning [79, 81, 83, 126, 133]. Four recent efforts have studied the problem of joint optimization of security or privacy and machine unlearning [45, 15, 143, 82].

certified removal is a certified-removal mechanism that applies a Newton step on the model parameters that largely remove the influence of the deleted data points [49]. GKT is a zero-shot machine unlearning algorithm that imposes the constraint that zero training data is available to the unlearning algorithms [20]. Two recent studies propose online machine unlearning methods for linear regression models [78] and linear support vector machine models [17], for further improving the efficiency of machine unlearning. The former adapts users' requests to delete their data before a specific time bar. The latter conducts only the task of variable support vector machine.

To our best knowledge, the common characteristics of the above machine unlearning methods is that they consist of two sequential operations: (1) Training: train a model on the complete training data and (2) Unlearning: generate an unlearning model from the former. The combination of two operations is computationally expensive when training complex models over large datasets. In addition, they often sequentially redo the unlearning operations one by one, when addressing a series of machine unlearning requests. This work is the first to execute one-time operation of simultaneous training and unlearning in advance for a series of machine unlearning requests, as long as the actual data removals are below the certified budget of data removals, by leveraging the theory of randomized smoothing and gradient quantization.

## A.2 Selection of Quantization Threshold

In Section 3, we have introduced the gradient quantization for the randomized smoothing for the certified data removals. Let $\lambda \geq 0$ be the quantization threshold.

$$Q(t) = Softmax([-|t - \lambda|, -|t|, -|t + \lambda|]) \tag{30}$$

where $Q(t)$ maps a gradient dimension $t$ to a three-dimensional vector $[-|t-\lambda|, -|t|, -|t+\lambda|]$, where each component denotes the similarity score between $t$ and $-\lambda$, 0, or $\lambda$. Therefore, all $T$ gradient dimensions are partitioned into three intervals: $(-\infty, -\lambda/2]$ that comes near to $-\lambda$, $[-\lambda/2, \lambda/2]$ that is closer to 0, and $[\lambda/2, \infty)$ that approaches $\lambda$. Each component in $Q(t)$ with the Softmax function also represents the probability of the gradient dimension $t$ belonging to classes -1, 0, or 1. Namely, the gradient quantization function $Q(t)$ will assign any gradient dimensions $t$ below $-\lambda/2$ to class -1, those above $\lambda/2$ to class 1, and the remaining dimensions to 0. The most probable class $c_A \in \{-1, 0, 1\}$ in $Q(t)$ is assigned to dimension $t$ as a final quantized gradient dimension.

In fact, the quantization threshold $\lambda \geq 0$ serves as a tradeoff hyperparameter to well balance the training performance and the certifiable radius regarding the data removals in our PCMU methods. On one hand, a large $\lambda$, say $\infty$, will cause most of the gradient dimensions to be assigned to class 0 and to be updated with zero. This may result in the failure of gradient and parameter updates and fail to train the model until convergence.

$$\int_{\frac{\lambda}{2}}^{\infty} \frac{1}{\sigma\sqrt{2\pi}} e^{-\frac{z^2}{2\sigma^2}} \, dz = \int_{-\infty}^{-\frac{\lambda}{2}} \frac{1}{\sigma\sqrt{2\pi}} e^{-\frac{z^2}{2\sigma^2}} \, dz << \int_{-\frac{\lambda}{2}}^{\frac{\lambda}{2}} \frac{1}{\sigma\sqrt{2\pi}} e^{-\frac{z^2}{2\sigma^2}} \, dz \tag{31}$$

On the other hand, a small $\lambda$, say 0, will make $\underline{p_A} = \overline{p_B}$, which bring small certified radius $R$ and thus result in the collapse of certified machine unlearning.

$$\int_{\frac{\lambda}{2}}^{\infty} \frac{1}{\sigma\sqrt{2\pi}} e^{-\frac{z^2}{2\sigma^2}} \, dz = \int_{-\infty}^{-\frac{\lambda}{2}} \frac{1}{\sigma\sqrt{2\pi}} e^{-\frac{z^2}{2\sigma^2}} \, dz >> \int_{-\frac{\lambda}{2}}^{\frac{\lambda}{2}} \frac{1}{\sigma\sqrt{2\pi}} e^{-\frac{z^2}{2\sigma^2}} \, dz \tag{32}$$

$$R = \frac{\sigma}{2} \left( \Phi^{-1} \left( \underline{p_A} \right) - \Phi^{-1} \left( \overline{p_B} \right) \right) \approx 0 \tag{33}$$

Therefore, we choose $\lambda = \sigma^2$ as the quantization threshold in this work, for ensuring the region $[-\lambda/2, \lambda/2]$ owns the largest area but does not dominate other two $(-\infty, -\lambda/2]$ and $[\lambda/2, \infty)$ in the PDF of Gaussian kernel.

### A.3 Algorithm

The following are the detailed descriptions of our PCMU method step by step: (1) Train the model in a usual manner with loss function $\mathcal{L}$, e.g., cross-entropy for image classification, and model parameter $w$; (2) Calculate the gradient $G(x,y) = \frac{\partial \mathcal{L}(x,y;w)}{\partial w}$ in Eq.(5) in the submission; (3) Compute the gradient average $\bar{G}$ in terms of the gradient $G(x_i, y_i)$ of each sample $(x_i, y_i)$ in Eq.(20) in the submission; (4) quantize each dimension $t$ ($t = 1, \cdots, T$) of the continuous gradient plus Gaussian noise $Q^t(\bar{G} + \varepsilon)$ in Eq.(21) over a discrete three-class space $\{-1, 0, 1\}$, for mimicking the classification in the randomized smoothing for certified robustness based on Eq.(6) in the submission; (5) Perform the randomized gradient smoothing for certified machine unlearning $S^{t\prime}(\bar{G}) = \underset{c \in \{-1,0,1\}}{\arg\max} \underset{\varepsilon \sim \mathcal{D}}{\mathbb{P}} (Q^t(\bar{G} + \varepsilon) = c)$ in Eq.(21) in the submission; (6) Derive the certified radius $R'$ in Eq.(24) and the certified budget $B'$ of data removals in Eq.(26) in the submission; (7) Integrate the model training for a specific learning task (e.g., image classification), randomized gradient smoothing, and gradient quantization into a unified framework for directly training a machine unlearning model with the data removal certificates as a guidance, for guaranteeing that the model parameters and gradients keep unchanged against the data removals within the certified budget, in terms of $w = w - \eta[S^{1\prime}(\bar{G}), \cdots, S^{T\prime}(\bar{G})]$ with smoothed and quantized gradients in Eq.(29) in the submission; and (8) Enter the next training round until convergence.

### A.4 Proof of Theorems

**Lemma 1.** *[Neyman-Pearson] Let $X$ and $Y$ be random variables in $\mathbb{R}^d$ with densities $\mu_x$ and $\mu_Y$. Let $h : \mathbb{R}^d \to \{0, 1\}$ be a random or deterministic function. Then:*

    *1. If $S = \{z \in \mathbb{R}^d : \frac{\mu_Y(z)}{\mu_X(z)} \le t\}$ for some $t > 0$ and $\mathbb{P}(h(X) = 1) \ge \mathbb{P}(X \in S)$, then $\mathbb{P}(h(Y) = 1) \ge \mathbb{P}(Y = S)$.*

    *2. If $S = \{z \in \mathbb{R}^d : \frac{\mu_Y(z)}{\mu_X(z)} \ge t\}$ for some $t > 0$ and $\mathbb{P}(h(X) = 1) \le \mathbb{P}(X \in S)$, then $\mathbb{P}(h(Y) = 1) \le \mathbb{P}(Y = S)$.*

*Proof. Please refer to the book [23] for detailed proof.*

**Lemma 2.** *[Neyman-Pearson for Gaussians with different means] Let $X \sim \mathcal{N}(x, \sigma^2 I)$ and $Y \sim \mathcal{N}(x + \delta, \sigma^2 I)$. Let $h : \mathbb{R}^d \to \{0, 1\}$ be any random or deterministic function. Then:*

    *1. If $S = \{z \in \mathbb{R}^d : \delta^T z \le \beta\}$ for some $\beta$ and $\mathbb{P}(h(X) = 1) \ge \mathbb{P}(X \in S)$, then $\mathbb{P}(h(Y) = 1) \ge \mathbb{P}(Y = S)$.*

    *2. If $S = \{z \in \mathbb{R}^d : \delta^T z \ge \beta\}$ for some $\beta > 0$ and $\mathbb{P}(h(X) = 1) \le \mathbb{P}(X \in S)$, then $\mathbb{P}(h(Y) = 1) \le \mathbb{P}(Y = S)$.*

*Proof. Please refer to the paper [23] for detailed proof.*

**Theorem 2.** *The error introduced by the Taylor expansion of $F_c^t(\hat{\bar{x}}, \hat{\bar{y}})$ at $(\bar{x}, \bar{y})$ is*

$$\epsilon \leq \sum_{j=0}^{\infty} ||\frac{L_{j+1}}{(j+1)!}|| \cdot ||\sigma M||^{j+1} \tag{34}$$

*where $L_j = \max_{k=1,\cdots,j} \frac{\partial^j h_i}{\partial x^k \partial y^{j-k}}$ and $M$ is the number of sampled points.*

*Proof.* When we have the Taylor expansion of $F_c^t(\hat{\bar{x}}, \hat{\bar{y}})$ at $(\bar{x}, \bar{y})$

$$\begin{aligned}
F_c^t(\hat{\bar{x}}, \hat{\bar{y}}) =& F_c^t(\bar{x}, \bar{y}) + \frac{\partial F_c^t}{\partial x}(\bar{x}, \bar{y})(\hat{\bar{x}} - \bar{x}) + \frac{\partial F_c^t}{\partial y}(\bar{x}, \bar{y})(\hat{\bar{y}} - \bar{y}) + \\
&+ \frac{1}{2}\{\frac{\partial^2 F_c^t(\bar{x}, \bar{y})}{\partial x^2}(\hat{\bar{x}} - \bar{x})^2 + 2\frac{\partial^2 F_c^t(\bar{x}, \bar{y})}{\partial x \partial y}(\hat{\bar{x}} - \bar{x})(\hat{\bar{y}} - \bar{y}) + \frac{\partial^2 F_c^t(\bar{x}, \bar{y})}{\partial y^2}(\hat{\bar{y}} - \bar{y})^2\} \\
&+ \cdots + O_j(\hat{\bar{x}}, \hat{\bar{y}})
\end{aligned} \tag{35}$$

*where $O_j(\hat{\bar{x}}, \hat{\bar{y}}) = \frac{1}{(j+1)!}\{\sum \cdots \sum \frac{\partial^k F_c^t(\xi)}{\partial x^k \partial y^{j+1-k}}(\hat{\bar{x}} - \bar{x})^k (\hat{\bar{y}} - \bar{y})^{j+1-k}\}$, $\xi \in ((\hat{\bar{x}}, \bar{x}), (\hat{\bar{y}}, \bar{y})), j = 3, \cdots, \infty$, and $k = 1, 2, 3$.*

*Then*

$$\begin{aligned}
\epsilon &= ||F_c^t(\hat{\bar{x}}, \hat{\bar{y}}) - F_c^t(\bar{x}, \bar{y})|| \\
&= ||\sum_{j=0}^{\infty} \frac{1}{(j+1)!}\{\sum \cdots \sum \frac{\partial^k F_c^t(\xi)}{\partial x^k \partial y^{j+1-k}}(\hat{\bar{x}} - \bar{x})^k (\hat{\bar{y}} - \bar{y})^{j+1-k}\}||
\end{aligned} \tag{36}$$

*Suppose that $F_c^t$ is at least second-order differentiable, and there exists Lipschitz constants $L_j = \max_{k=1,\cdots,j} \frac{\partial^j h_i}{\partial x^k \partial y^{j-k}}$ in each function, then*

$$\epsilon_2 \leq \sum_{j=0}^{\infty} ||\frac{L_{j+1}}{(j+1)!}|| \cdot ||\varepsilon_x||^k \cdot ||\varepsilon_y||^{j+1-k} \leq \sum_{j=0}^{\infty} ||\frac{L_{j+1}}{(j+1)!}|| \cdot ||\sigma M||^{j+1} \tag{37}$$

**Theorem 3.** *Let $\varepsilon \sim \mathcal{D} = \mathcal{N}(0, \sigma^2 I)$ and $\tilde{S}^t(\bar{x}) = \underset{c \in \{-1,0,1\}}{\arg\max} \underset{\varepsilon \sim \mathcal{D}}{\mathbb{P}}(\tilde{F}^t(\bar{x} + \varepsilon) = c)$. Suppose that for a specific $\bar{x} \in \mathbb{R}^d$, there exist $c_A \in \{-1, 0, 1\}$ and $\underline{p_A}, \overline{p_B} \in [0, 1]$ such that:*

$$\mathbb{P}\left(\tilde{F}^t(\bar{x} + \varepsilon) = c_A\right) \geq \underline{p_A} \geq \overline{p_B} \geq \max_{c \neq c_A} \mathbb{P}(\tilde{F}^t(\bar{x} + \varepsilon) = c) \tag{38}$$

*Then $\tilde{S}^t(\bar{x} + \delta) = c_A$ for all $||\delta||_2 < R$, where*

$$R = \frac{\sigma}{2}\left(\Phi^{-1}\left(\underline{p_A}\right) - \Phi^{-1}\left(\overline{p_B}\right)\right) \tag{39}$$

*where $\Phi^{-1}$ is the inverse of the standard Gaussian CDF.*

*Proof.* Notice that the correlation between the data $\bar{x}$ and its classes $\bar{y}$ is fixed for a given dataset. We denote this correlation as $\bar{y} = H(\bar{x})$ where $H : \mathbb{R}^d \mapsto C$. Thus, the original $F^t(\bar{x}, \bar{y})$ can be rewritten as an equivalent one $\tilde{F}^t(\bar{x})$.

$$\tilde{F}^t(\bar{x}) = F^t(\bar{x}, H(\bar{x})) = F^t(\bar{x}, \bar{y}) \tag{40}$$

Based on the definition of $\tilde{S}^t(\bar{x})$, to prove $\tilde{S}^t(\bar{x} + \delta) = c_A$ for all $\|\delta\|_2 < R$, we first need to demonstrate

$$\mathbb{P}(\tilde{F}^t(\bar{x} + \delta + \varepsilon) = c_A) > \max_{c_B \neq c_A} \mathbb{P}(\tilde{F}^t(\bar{x} + \delta + \varepsilon) = c_B) \tag{41}$$

For ease of presentation, two random variables are defined as follows.

$$u = \bar{x} + \varepsilon = \mathcal{N}(\bar{x}, \sigma^2 I) \tag{42}$$

$$v = \bar{x} + \delta + \varepsilon = \mathcal{N}(\bar{x} + \delta, \sigma^2 I) \tag{43}$$

Based on the given condition in Eq.(38), we know

$$\mathbb{P}(\tilde{F}^t(u)) = c_A) \geq \underline{p_A} \tag{44}$$

$$\mathbb{P}(\tilde{F}^t(v) = c_B) \leq \overline{p_B} \tag{45}$$

We define two half-spaces as follows.

$$X = \{U : \delta^T(U - \bar{x}) \leq \sigma\|\delta\|\Phi^{-1}(\underline{p_A})\} \tag{46}$$

$$Y = \{U : \delta^T(U - \bar{x}) \geq \sigma\|\delta\|\Phi^{-1}(1 - \overline{p_B})\} \tag{47}$$

Now, we have

$$\begin{aligned}
\mathbb{P}(u \in X) &= \mathbb{P}\left(\delta^T(u - \bar{x}) \leq \sigma\|\delta\|\Phi^{-1}\left(\underline{p_A}\right)\right) = \mathbb{P}\left(\delta^T\mathcal{N}\left(0, \sigma^2 I\right) \leq \sigma\|\delta\|\Phi^{-1}\left(\underline{p_A}\right)\right) \\
&= \mathbb{P}\left(\sigma\|\delta\|z \leq \sigma\|\delta\|\Phi^{-1}\left(\underline{p_A}\right)\right) = \mathbb{P}\left(z \leq \Phi^{-1}\left(\underline{p_A}\right)\right) = \Phi\left(\Phi^{-1}\left(\underline{p_A}\right)\right) \\
&= \underline{p_A}
\end{aligned} \tag{48}$$

where $z \sim \mathcal{N}(0, 1)$.

By combining Eq.(44) and Eq.(48), we have

$$\mathbb{P}(\tilde{F}^t(u)) = c_A) \geq \mathbb{P}(u \in X) \tag{49}$$

By applying Lemma 2 with $h(u) = \mathbf{1}[\tilde{F}^t(u) = c_A]$, we obtain

$$\mathbb{P}(\tilde{F}^t(v)) = c_A) \geq \mathbb{P}(v \in X) \tag{50}$$

Similarly, we have

$$\begin{aligned}
\mathbb{P}(u \in Y) &= \mathbb{P}\left(\delta^T(u - \bar{x}) \geq \sigma\|\delta\|\Phi^{-1}\left(1 - \overline{p_B}\right)\right) = \mathbb{P}\left(\delta^T\mathcal{N}\left(0, \sigma^2 I\right) \geq \sigma\|\delta\|\Phi^{-1}\left(1 - \overline{p_B}\right)\right) \\
&= \mathbb{P}\left(\sigma\|\delta\|z \geq \sigma\|\delta\|\Phi^{-1}\left(1 - \overline{p_B}\right)\right) = \mathbb{P}\left(z \geq \Phi^{-1}\left(1 - \overline{p_B}\right)\right) = 1 - \Phi\left(\Phi^{-1}\left(1 - \overline{p_B}\right)\right) \\
&= \overline{p_B}
\end{aligned} \tag{51}$$

Again, we get

$$\mathbb{P}(\tilde{F}^t(u)) = c_B) \leq \mathbb{P}(u \in Y) \tag{52}$$

*and*

$$\mathbb{P}(\tilde{F}^t(v)) = c_B) \leq \mathbb{P}(v \in Y) \tag{53}$$

*Notice that our proof objective $\tilde{S}^t(\bar{x} + \delta) = c_A$ is equivalent to the following inequality.*

$$\mathbb{P}(\tilde{F}^t(v)) = c_A) > \mathbb{P}(\tilde{F}^t(v)) = c_B) \tag{54}$$

*If we can prove $\mathbb{P}(v \in X) > \mathbb{P}(v \in Y)$, then we can obtain*

$$\mathbb{P}(\tilde{F}^t(v)) = c_A) \geq \mathbb{P}(v \in X) > \mathbb{P}(v \in Y) \geq \mathbb{P}(\tilde{F}^t(v)) = c_B) \tag{55}$$

*Now, we calculate the condition satisfying $\mathbb{P}(v \in X) > \mathbb{P}(v \in Y)$.*

$$\mathbb{P}(v \in X) = \Phi(\Phi^{-1}(\underline{p_A}) - \frac{||\delta||}{\sigma}) \tag{56}$$

*and*

$$\mathbb{P}(v \in Y) = \Phi(\Phi^{-1}(\overline{p_B}) + \frac{||\delta||}{\sigma}) \tag{57}$$

$\mathbb{P}(v \in X) > \mathbb{P}(v \in Y)$ *is satisfied if and only if*

$$||\delta|| < \frac{\sigma}{2}(\Phi^{-1}(\underline{p_A}) - \Phi^{-1}(\overline{p_B})) \tag{58}$$

*Therefore, the proof is concluded.*

**Theorem 4.** *Let $R$ be the certified radius of $\bar{x} \in \mathbb{R}^d$ based on $\tilde{S}^t(\bar{x}) = \underset{c \in \{-1,0,1\}}{\operatorname{argmax}} \underset{\varepsilon \sim \mathcal{D}}{\mathbb{P}}(\tilde{F}^t(\bar{x}+\varepsilon) = c)$, then the certified budget of data removal is*

$$B \leq N - \frac{9d\sigma^2}{R^2} \tag{59}$$

*where $N$ is the number of data samples on the entire training data and $B$ is the maximally allowed number of data samples escaped from the training data.*

*Proof. Without loss of generality, suppose that $x \sim \mathcal{N}(\bar{x}, \sigma^2 I)$. The complete training data $D$ is partitioned into two subsets: the forgotten data $D_f \subseteq D$ and the remembered data $D_r \subseteq D$ ($D = D_f \cup D_r$, $D_f \cap D_r = \emptyset$). Let $\{x_1, \cdots, x_B\}$ be the data in $D_f$, $\{x_{B+1}, \cdots, x_N\}$ be the data in $D_r$, and $\bar{x}_r$ be the average of all data samples in $D_r$.*

$$\bar{x}_r = \frac{1}{N-B} \sum_{x_i \in D_r} x_i \tag{60}$$

*We calculate the expectation and variance of $\bar{x}_r$ about possible escape situations.*

$$\mathbf{E}(\bar{x}_r) = \frac{1}{N-B} \sum_{i=B+1}^{N} x_i = \bar{x}_r \tag{61}$$

$$\mathbf{Var}(\bar{x}_r) = \mathbf{Var}(\frac{\sum_{i=B+1}^{N} x_i}{N-B}) = \frac{1}{(N-B)^2} \mathbf{Var}(\sum_{i=B+1}^{N} x_i) = \frac{\sigma^2 I}{N-B} \tag{62}$$

*Thus, the data samples $x_{B+1}, \cdots, x_N$ in $D_r$ follow $\mathcal{N}(\bar{x}_r, \frac{\sigma^2 I}{N-B})$.*

*If we want to guarantee the forgotten data (i.e., the data removals) within the certified radius $R$, then we need to ensure*

$$\mathbb{P}\{||\bar{x} - \bar{x}_r|| \leq R\} = 99.73\% \approx 1, \tag{63}$$

*in terms of the three-sigma rule. $\bar{x}$ is the average of all data samples in the entire training data.*

*Thus, we have*

$$R \geq 3||\frac{\sigma I}{\sqrt{N - B}}|| \tag{64}$$

*Therefore, we obtain*

$$B \leq N - \frac{9d\sigma^2}{R^2} \tag{65}$$

*By combining Eq.(39) and Eq.(65) together, we further get*

$$B \leq N - \frac{36d}{(\Phi^{-1}(\underline{p_A}) - \Phi^{-1}(\overline{p_B}))^2} \tag{66}$$

**Theorem 5.** *Let $R$ and $R'$ be the certified radii of the above two algorithms respectively and $L$ be the Lipschitz constant of gradient $G(x, y) \in \mathbb{R}^T$, then*

$$R \geq \frac{\sqrt{T}}{L}R' \tag{67}$$

*By combining Theorems 4 and 5 together, we derive the certified budget $B'$ of data removal from $R'$.*

$$B' \leq N - \frac{36dL^2}{T(\Phi^{-1}(\underline{p_{A'}}) - \Phi^{-1}(\overline{p_{B'}}))^2} \tag{68}$$

*Proof. Let $G(x, y) \in \mathbb{R}^T$ be the gradient of a machine learning model.*

$$G(x, y) = \frac{\partial \mathcal{L}(x, y; w)}{\partial w} \tag{69}$$

*Let $G^t(x, y)$ be the $t^{th}$ $(t = 1, \cdots, T)$ dimension of the gradient $G(x, y)$, $\tilde{G}^t(\bar{x}) = G^t(\bar{x}, H(\bar{x})) = G^t(\bar{x}, \bar{y})$, and $\tilde{Q}^t(\tilde{G}^t(\bar{x})) = Q^t(G^t(\bar{x}, \bar{y}))$. We use $\tilde{Q}_c^t(\tilde{G}^t(\bar{x}))$ to represent the $c^{th}$ $(c \in \{-1, 0, 1\})$ component of $\tilde{Q}^t(\tilde{G}^t(\bar{x}))$.*

*For the randomized data smoothing and gradient quantization method, we have*

$$\underline{p_A} = \int_{\bar{x}} \mathbb{P}(\tilde{F}_c^t(\bar{x}))d\bar{x} = \int_{x_i} \mathbb{P}(\tilde{F}_c^t(\mathbf{E}(x_i)))dx_i = \int_{x_i} \mathbb{P}(\tilde{Q}_c^t(\tilde{G}^t(\mathbf{E}(x_i))))dx_i, \ c = c_A \tag{70}$$

$$\overline{p_B} = \int_{\bar{x}} \mathbb{P}(\tilde{F}_c^t(\bar{x}))d\bar{x} = \int_{x_i} \mathbb{P}(\tilde{F}_c^t(\mathbf{E}(x_i)))dx_i = \int_{x_i} \mathbb{P}(\tilde{Q}_c^t(\tilde{G}^t(\mathbf{E}(x_i))))dx_i, \ c \neq c_A \tag{71}$$

$$R = \frac{\sigma}{2}\left(\Phi^{-1}\left(\underline{p_A}\right) - \Phi^{-1}\left(\overline{p_B}\right)\right) \tag{72}$$

*For the randomized gradient smoothing and quantization approach, we have*

$$\underline{p'_A} = \int_{\bar{G}^t} \mathbb{P}(Q_c^t(\bar{G}^t))d\bar{G}^t = \int_{x_i} \int_{y_i} \mathbb{P}(Q_c^t(\mathbf{E}(G^t(x_i, y_i))))dx_i dy_i$$

$$= \int_{x_i} \mathbb{P}(\tilde{Q}_c^t(\mathbf{E}(\tilde{G}^t(x_i))))dx_i, \ c = c_A \tag{73}$$

$$\overline{p'_B} = \int_{\bar{G}^t} \mathbb{P}(Q_c^t(\bar{G}^t))d\bar{G}^t = \int_{x_i} \int_{y_i} \mathbb{P}(Q_c^t(\mathbf{E}(G^t(x_i, y_i))))dx_i dy_i$$

$$= \int_{x_i} \mathbb{P}(\tilde{Q}_c^t(\mathbf{E}(\tilde{G}^t(x_i))))dx_i, \ c \neq c_A \tag{74}$$

$$R' = \frac{\sigma}{2} \left( \Phi^{-1}\left(\underline{p'_A}\right) - \Phi^{-1}\left(\overline{p'_B}\right) \right) \tag{75}$$

*Let L be the Lipschitz constant of gradient $G(x, y)$, for any $\delta > 0$, we have*

$$L \cdot ||\delta|| \geq ||\mathbf{E}(\tilde{G}^t(x_i + \delta)) - \mathbf{E}(\tilde{G}^t(x_i))|| = \sqrt{T}R' \tag{76}$$

*Then the minimum change in $\bar{x}$ is*

$$\min ||\delta|| = \frac{\sqrt{T}}{L}R' \tag{77}$$

*This implies*

$$R \geq \frac{\sqrt{T}}{L}R' \tag{78}$$

*By combining Eq.(65) and Eq.(78), we obtain the certified budget $B'$ of data removals in the randomized gradient smoothing and quantization approach.*

$$B' \leq N - \frac{9d\sigma^2}{(\frac{\sqrt{T}}{L}R')^2}$$

$$\leq N - \frac{36dL^2}{T(\Phi^{-1}(\underline{p_A}') - \Phi^{-1}(\overline{p_B}'))^2} \tag{79}$$

**Theorem 6.** *Let $S^{t\prime}(\bar{G})$ be the randomized gradient smoothing for certified machine unlearning on gradient quantization, $L$, $L_1$, and $L_2$ be the Lipschitz constants of $G$, $Q^t$, and $S^{t\prime}$ respectively, i.e.,*

$$||\nabla S^{t\prime}(a) - \nabla S^{t\prime}(b)||_2 \leq L_2 L_1 L ||a - b||_2 \ for \ any \ a, b \tag{80}$$

*If we run gradient descent for $k$ iterations with a fixed step size $s \leq \frac{1}{L_2 L_1 L}$, it will yield a solution $S^{t\prime(k)}$ which satisfies*

$$S^{t\prime}(q^{(k)}) - S^{t\prime}(q^*) \leq \frac{||q^{(0)} - q^*||_2^2}{2sk} \tag{81}$$

*where $S^{t\prime}(q^{(0)})$ is the initial solution and $S^{t\prime}(q^*)$ is the local optimal solution.*

*This means that gradient descent is guaranteed to converge and that it converges with rate $\mathcal{O}(1/k)$.*

*Proof. Suppose that $S^{t\prime}$ is local convex and differentiable. Let $q^{(s)}$ be the gradient $S^{t\prime}(\bar{G})$ with the randomized gradient smoothing and gradient quantization at the $s^{th}$ training iteration and $q^*$ be the local optimal solution of $q^{(s)}$.*

*For any $a, b$ in the local convex domain of $S^{t\prime}$, we have*

$$
\begin{aligned}
S^{t\prime}(b) &\leq S^{t\prime}(a) + \nabla S^{t\prime}(a)^T (b-a) + \frac{1}{2} S^{t\prime}(a) ||b-a||_2^2 \\
&\leq S^{t\prime}(a) + \nabla S^{t\prime}(a)^T (b-a) + \frac{1}{2} L_2 L_1 L ||b-a||_2^2
\end{aligned}
\tag{82}
$$

*We plug in the gradient descent update by letting $b = a^{(+)} = a - t\nabla F(a)$.*

$$
\begin{aligned}
S^{t\prime}(a^{(+)}) &\leq S^{t\prime}(a) + \nabla S^{t\prime}(a)^T (a^{(+)} - a) + \frac{1}{2} L_2 L_1 L ||a^{(+)} - a||_2^2 \\
&= S^{t\prime}(a) + \nabla S^{t\prime}(a)^T (a - s\nabla S^{t\prime}(a) - a) + \frac{1}{2} L_2 L_1 L ||a - s\nabla S^{t\prime}(a) - a||_2^2 \\
&= S^{t\prime}(a) - \nabla S^{t\prime}(a)^T s\nabla S^{t\prime}(a) + \frac{1}{2} L_2 L_1 L ||s\nabla S^{t\prime}(a)||_2^2 \\
&= S^{t\prime}(a) - s||\nabla S^{t\prime}(a)||_2^2 + \frac{1}{2} L_2 L_1 L s^2 ||\nabla S^{t\prime}(a)||_2^2 \\
&= S^{t\prime}(a) - (1 - \frac{1}{2} L_2 L_1 L s) s ||\nabla S^{t\prime}(a)||_2^2.
\end{aligned}
\tag{83}
$$

*Based on $s \leq \frac{1}{L_2 L_1 L}$, we have*

$$
-(1 - \frac{1}{2} L_2 L_1 L s) = \frac{1}{2} L_2 L_1 L s - 1 \leq -\frac{1}{2}
\tag{84}
$$

*By plugging Eq.(84) into Eq.(83), we obtain*

$$
S^{t\prime}(a^{(+)}) \leq S^{t\prime}(a) - \frac{1}{2} s ||\nabla S^{t\prime}(a)||_2^2
\tag{85}
$$

*Since $\frac{1}{2} s ||\nabla S^{t\prime}(a)||_2^2$ is always non-negative, the inequality in Eq.(85) implies that the objective function value strictly decreases with the iteration of gradient descent until it reaches the local optimal value $S^{t\prime}(a) = S^{t\prime}(a^*)$.*

*Now we need to bound the objective value at the next iteration, $S^{t\prime}(a^{(+)})$, in terms of the local optimal objective value $S^{t\prime}(a^*)$.*

*Since $S^{t\prime}$ is local convex, we have*

$$
S^{t\prime}(a^*) \geq S^{t\prime}(a) + \nabla S^{t\prime}(a)^T (a^* - a)
\tag{86}
$$

$$
S^{t\prime}(a) \geq S^{t\prime}(a^*) + \nabla S^{t\prime}(a)^T (a - a^*)
\tag{87}
$$

*According to (84), we obtain*

$$
S^{t\prime}(a^{(+)}) \leq S^{t\prime}(a^*) + \nabla S^{t\prime}(a)^T (a - a^*) - \frac{s}{2} ||\nabla S^{t\prime}(a)||_2^2
\tag{88}
$$

*Notice that*

$$
||a - s\nabla S^{t\prime}(a) - a^*||_2^2 = ||a - a^*||_2^2 - 2s\nabla S^{t\prime}(a)^T (a - a^*) + s^2 ||\nabla S^{t\prime}(a)||_2^2
\tag{89}
$$

*Thus, we have*

$$S^{t\prime}(a^{(+)}) - S^{t\prime}(a^*) \leq \frac{1}{2s}(||a - a^*||_2^2 - ||a - s\nabla S^{t\prime}(a) - a^*||_2^2) \tag{90}$$

*Notice that $a^{(+)} = a - s\nabla S^{t\prime}(a)$, by plugging this into (90), we get*

$$S^{t\prime}(a^{(+)}) - S^{t\prime}(a^*) \leq \frac{1}{2s}(||a - a^*||_2^2 - ||a^{(+)} - a^*||_2^2) \tag{91}$$

*By aggregating the terms at all iterations, we have*

$$
\begin{aligned}
\sum_{i=1}^{k} S^{t\prime}(a^{(k)} - S^{t\prime}(a^*)) &\leq \sum_{i=1}^{k} \frac{1}{2s}(||a^{(i-1)} - a^*||_2^2 - ||a^{(i)} - a^*||_2^2) \\
&= \frac{1}{2s}(||a^{(0)} - a^*||_2^2 - ||a^{(k)} - a^*||_2^2) \\
&\leq \frac{1}{2s}(||a^{(0)} - a^*||_2^2)
\end{aligned}
\tag{92}
$$

*Finally, since the function $S^{t\prime}$ keeps decreasing at each iteration, we can conclude*

$$S^{t\prime}(a^{(k)}) - S^{t\prime}(a^*) \leq \frac{1}{k}\sum_{i=1}^{k} S^{t\prime}(a^{(i)}) - S^{t\prime}(a^*) \leq \frac{||a^{(0)} - a^*||_2^2}{2sk} \tag{93}$$

*By replacing $a$ with $q$, the proof is concluded.*

### A.5  Additional Experiments

**Machine unlearning performance and running time with varying ratios of data removal.** Tables 5-15 exhibit the classification accuracy, errors, training time, and unlearning time obtained by eleven machine unlearning approaches by varying the ratio of unlearning request / data removal between 2% and 20% on three datasets of Fashion-MNIST, CIFAR-10, and SVHN respectively. Similar trends are observed for the comparison of machine unlearning effectiveness and efficiency in these figures: our PCMU method achieves the smallest absolute performance difference with the Retrain model, regarding $Accuracy$ (<1%), $Error_t$ (<1%), $Error_r$ (<4%), and $Error_f$ (<1%) on three datasets respectively. Our PCMU method achieves better efficiency than most baseline methods, except DeltaGrad and Unrolling SGD. Our PCMU method performs one-time operation of simultaneous training and unlearning when addressing a series of machine unlearning requests. Thus, our PCMU method has only one running time for multiple unlearning requests, e.g., 2,566 seconds on SVHN for all five unlearning results (2%, 3%, 5%, 10%, and 20%). However, other baselines need to sequentially handle these machine unlearning requests one by one. Therefore, they have multiple running time for varying ratios of data removals, e.g., 1,176, 1,195, 1,231, 1,280, and 1,371 seconds achieved by Unrolling SGD on SVHN for five unlearning requests (2%, 3%, 5%, 10%, and 20%) respectively. The above experiment results demonstrate that PCMU is effective as well as efficient for addressing the machine unlearning problem. This advantage is very important for entitling data owners to the right to have their private data removed from trained complex models at their requests in a timely and cost-efficient manner in privacy-critical applications that usually require near-zero tolerance of data leaking.

Table 5: Performance with 5% data removal and CNN on Fashion-MNIST

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

**Machine unlearning performance and running time with different smoothing techniques.** Tables 16 and 17 evaluate the impact of different smoothing techniques on our proposed prompt certified machine unlearning method, PCMU, over two popular image classification datasets: Fashion-MNIST and CIFAR-10. We replace the randomized smoothing component in the original PCMU model with Laplacian smoothing and uniform smoothing respectively. It is observed that two PCMU variants with Laplacian smoothing and uniform smoothing achieve the close performance to the original PCMU model with randomized smoothing, showing the generality of PCMU to the machine unlearning. Compared with the results achieved by the nine state-of-the-art baselines in Tables 2 and 4, two PCMU variants still substantially outperform the performance of other baselines in most experiments. We will include all the experiment results in this rebuttal into the submission.

Table 16: Performance with 20% data removal and CNN on Fashion-MNIST

| Metric | Performance | | | | Runtime (s) | | |
|---|---|---|---|---|---|---|---|
| | $Accuracy$ | $Error_t$ | $Error_r$ | $Error_f$ | Training | Unlearning | Total |
| Retrain | 88.21 | 11.79 | 9.75 | 11.76 | 687 | 561 | 1,248 |
| PCMU | **88.34** | **11.66** | **10.25** | **11.47** | 802 | 0 | 802 |
| PCMU+Laplacian Smoothing | 86.16 | 13.84 | 12.29 | 13.20 | 773 | 0 | 773 |
| PCMU+Uniform Smoothing | 86.86 | 13.14 | 10.90 | 12.77 | 804 | 0 | 804 |

Table 17: Performance with 20% data removal and LeNet on CIFAR-10

| Metric | Performance | | | | Runtime (s) | | |
|---|---|---|---|---|---|---|---|
| | $Accuracy$ | $Error_t$ | $Error_r$ | $Error_f$ | Training | Unlearning | Total |
| Retrain | 63.29 | 36.71 | 24.59 | 36.89 | 846 | 673 | 1,519 |
| PCMU | **64.33** | **35.67** | **25.18** | **35.32** | 903 | 0 | 903 |
| PCMU+Laplacian Smoothing | 61.77 | 38.23 | 29.41 | 38.30 | 928 | 0 | 928 |
| PCMU+Uniform Smoothing | 62.17 | 37.83 | 28.06 | 38.60 | 936 | 0 | 936 |

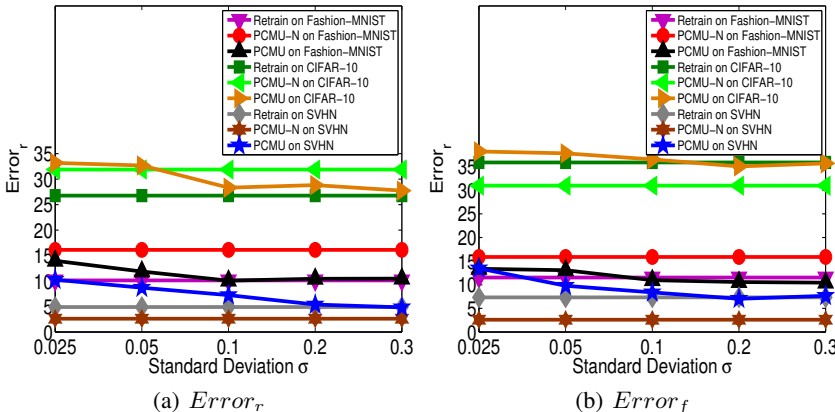

(a) $Error_r$        (b) $Error_f$

Figure 3: Errors with varying standard deviation on three datasets

**Machine unlearning performance and running time based on removals of class samples.** In the submission, all the formulae, methods, and theories do not have assumptions or restrictions regarding the distribution of data removals. Thus, our proposed prompt certified machine unlearning method, PCMU, is able to work on the case of removals of class samples, as long as the actual data removals are below the certified budget of data removals. In fact, the removals of arbitrary data samples in our experiments is more general than the removals of class samples.

In order to validate the performance of our PCMU for this special machine unlearning problem, we randomly choose one class and remove all samples from this class over CIFAR-10. Table 18 exhibits the corresponding experiment results, which demonstrate that PCMU is also effective as well as efficient for addressing this special machine unlearning problem.

Table 18: Performance with data removal of samples from one class and LeNet on CIFAR-10

| Metric | Performance | | | | Runtime (s) | | |
|---|---|---|---|---|---|---|---|
| | $Accuracy$ | $Error_t$ | $Error_r$ | $Error_f$ | Training | Unlearning | Total |
| Retrain | 58.91 | 41.09 | 28.23 | 100 | 859 | 738 | 1,597 |
| Fisher | 62.28 | 37.72 | 34.28 | 29.51 | 1,459 | 1,490 | 2,949 |
| certified removal | 38.40 | 61.61 | 61.73 | 60.32 | **886** | 218 | 1,104 |
| NTK | 62.71 | 37.29 | 32.47 | 42.20 | 1,459 | 1,353 | 2,812 |
| MCMC unlearning | 57.13 | 42.87 | 16.44 | 49.52 | 1,565 | 734 | 2,299 |
| PCMU | **60.15** | **39.85** | **26.51** | **72.18** | 926 | **0** | **926** |

## A.6 Parameter Sensitivity

In this section, we conduct more experiments to validate the sensitivity of various parameters in our PCMU method for the certified machine unlearning task.

**Impact of standard deviation.** Figure 3 (a) and (b) measure the effect of standard deviation of the Gaussian distribution in the randomized gradient smoothing for machine unlearning on $Error_r$ and $Error_f$ by varying $\sigma$ from 0.025 to 0.3. The error scores achieved by the Retrain and PCMU-N models keep unchanged with varying $\sigma$. We have observed similar results in these two figures: The error curves by PCMU initially decrease quickly and then become stable when $\sigma$ continuously increases. A suitable $\sigma$ can help utilize the randomized gradient smoothing and quantization for directly training a certfied machine unlearning model in advance. A too large $\sigma$ beyond some thresholds does not affect the performance of machine unlearning any more.

**Influence of training sample percentage.** Figure 4 (a) shows the influence of training sample percentage in our PCMU model by varying it from 20% to 100%. We make the observations on the quality by three machine unlearning methods. (1) The accuracy by our PCMU model is very close to that of the Retrain method in most experiments. (2) The performance curves keep increasing when the number of training samples increases. (3) PCMU outperforms PCMU-N in most tests with the smallest accuracy difference with the Retrain method. When there are many training samples available ($\geq 40\%$), the quality improvement by PCMU is obvious. A reasonable explanation is more

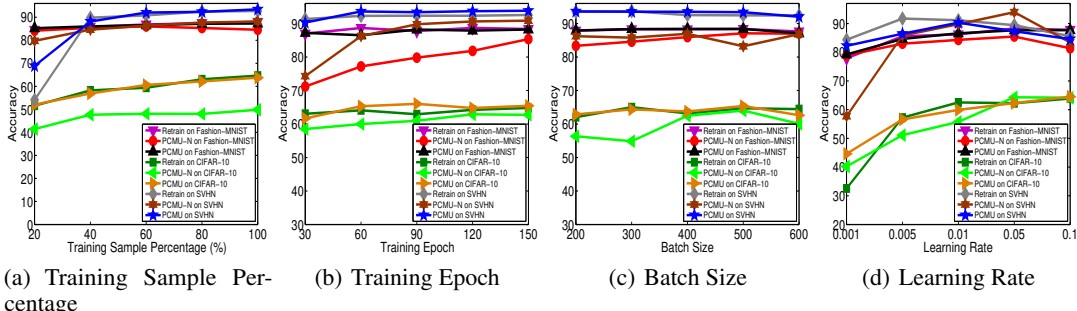

| (a) Training Sample Percentage | (b) Training Epoch | (c) Batch Size | (d) Learning Rate |

Figure 4: Performance with varying Parameters on three datasets

training data makes PCMU be more resilient to machine unlearning under suitable ratios of data removals.

**Impact of training epochs.** Figure 4 (b) exhibits the sensitivity of training epochs of our PCMU model by varying them from 30 and 150. As we can see, the performance curves continuously increase with increasing training epochs. This is consistent with the fact that more training epochs makes the image classification models be resilient to machine unlearning under suitable ratios of data removals. It is observed that the accuracy scores oscillate within the range of 3.1% on three datasets.

**Sensitivity of batch size.** Figure 4 (c) exhibits the sensitivity of batch size of machine unlearning models in our PCMU model by varying them from 200 and 600. It is observed that the performance curves keep relatively stable when we continuously change the batch size. This demonstrates that our PCMU method is insensitive to the batch size of machine unlearning. No matter what the batch size is, our PCMU method can always achieve the superior performance in all tests, showing the effectiveness of our PCMU method to the machine unlearning.

**Influence of learning rates.** Figure 4 (d) shows the influence of learning rate in our PCMU model by varying it from 0.001 to 0.1. We have observed that the accuracy initially raises when the learning rate increases. Intuitively, a large learning rate can help the algorithm quickly find the optimal solution and thus help improve the quality of machine unlearning. Later on, the performance curves decrease quickly when the learning rate continuously increases. A reasonable explanation is that a too large learning rate may miss the optimal solution with large step size in the search process. Thus, it is important to determine the optimal learning rate for the machine unlearning.

### A.7 Experimental Details

**Environment.** The experiments were conducted on a compute server running on Red Hat Enterprise Linux 7.2 with 2 CPUs of Intel Xeon E5-2650 v4 (at 2.66 GHz) and 8 GPUs of NVIDIA GeForce GTX 2080 Ti (with 11GB of GDDR6 on a 352-bit memory bus and memory bandwidth in the neighborhood of 620GB/s), 256GB of RAM, and 1TB of HDD. Overall, the experiments took about 3 days in a shared resource setting. We expect that a consumer-grade single-GPU machine (e.g., with a 2080 Ti GPU) could complete the full set of experiments in around 4-5 days, if its full resources were dedicated. The codes were implemented in Python 3.7.3 and PyTorch 1.0.14. We also employ Numpy 1.16.4 and Scipy 1.3.0 in the implementation. Since the datasets used are all public datasets and our methodologies and the hyperparameter settings are explicitly described in Section 3, 4, 5, and A.7, our codes and experiments can be easily reproduced on top of a GPU server.

**Training.** We study image classification networks on three standard image datasets: Fashion-MNIST [2], CIFAR-10 [3], and SVHN [4]. The above three image datasets are all public datasets, which allow researchers to use for non-commercial research and educational purposes. We use 60,000 examples as training data and 10,000 examples as test data for Fashion-MNIST. We train the machine unlearning model on the CIFAR-10 training set and test it on the CIFAR-10 test set. We use 73,257 digits as training data and 26,032 digits as test data for SVHN. We train a convolutional neural network (CNN) on Fashion-MNIST for clothing classification. We train LeNet over CIFAR-10 for

---

[2]https://github.com/zalandoresearch/fashion-mnist

[3]https://www.cs.toronto.edu/∼kriz/cifar.html

[4]http://ufldl.stanford.edu/housenumbers/

image classification. We apply the ResNet-18 architecture on SVHN for street view house number identification. The neural networks are trained with Kaiming initialization [52] using SGD for 120 epochs with an initial learning rate of 0.05 and batch size 500. The learning rate is decayed by a factor of 0.1 at 1/2 and 3/4 of the total number of epochs. In addition, we run each experiment for 3 trials for obtaining more stable results.

**Implementation.** For nine state-of-the-art machine unlearning models of Fisher [5], certified removal [6], DeltaGrad [7], NTK [8], Unrolling SGD [9], SISA [10], Adaptive Unlearning [11], FedEraser [12], and MCMC unlearning [13], we utilized the same model architecture as the official open-source implementation and default parameter settings provided by the original authors for machine unlearning in all experiments. All hyperparameters are standard values from reference codes or prior works. We validate the performance of different machine unlearning methods with a range of ratio of data removals $\{5\%, 8\%, 10\%, 15\%, 20\%\}$. All models were trained for 120 epochs, with a batch size of 500, and a learning rate of 0.05. The above open-source codes from the GitHub are licensed under the MIT License, which only requires preservation of copyright and license notices and includes the permissions of commercial use, modification, distribution, and private use.

For our PCMU model, we performed hyperparameter selection by performing a parameter sweep on standard deviation $\sigma \in \{0.025, 0.05, 0.1, 0.2, 0.3, 0.5, 1\}$ in the Gaussian distribution, quantization threshold $\lambda \in \{\sigma^2/4, \sigma^2/2, \sigma^2, 2\sigma^2, 4\sigma^2\}$, ratio of data removals $\{5\%, 8\%, 10\%, 15\%, 20\%\}$, training epochs of the machine unlearning model $\in \{30, 60, 90, 120, 150\}$, batch size for training the model $\in \{200, 300, 400, 500, 600\}$, and learning rate $\in \{0.001, 0.005, 0.01, 0.05, 0.1, 0.5\}$. We select the best parameters over 50 epochs of training and evaluate the model at test time.

**Hyperparameter settings.**

Unless otherwise explicitly stated, we used the following default parameter settings in the experiments.

Table 19: Hyperparameter Settings

| Parameter | Value |
|---|---|
| Training data on Fashion-MNIST | 60,000 |
| Test data ratio on Fashion-MNIST | 10,000 |
| Training data on CIFAR-10 | 50,000 |
| Test data on CIFAR-10 | 10,000 |
| Training data on SVHN | 73,257 |
| Test data on SVHN | 26,032 |
| Standard deviation $\sigma$ in the Gaussian distribution | 0.1 |
| Quantization threshold $\lambda$ | $\sigma^2$ |
| Ratio of data removals | 20% |
| Training epochs of the machine unlearning model | 120 |
| Batch size for training the model | 500 |
| Learning rate | 0.05 |

## A.8  Potential Negative Societal Impacts and Limitations

In this work, the three image datasets are all open-released datasets [138, 66, 95], which allow researchers to use for non-commercial research and educational purposes. These three datasets are widely used in training/evaluating the image classification. All baseline codes are open-accessed resources that are from the GitHub and licensed under the MIT License, which only requires

---

[5]https://github.com/AdityaGolatkar/SelectiveForgetting

[6]https://github.com/facebookresearch/certified-removal

[7]https://github.com/thuwuyinjun/DeltaGrad

[8]https://github.com/AdityaGolatkar/SelectiveForgetting

[9]https://github.com/cleverhans-lab/unrolling-sgd

[10]https://github.com/cleverhans-lab/machine-unlearning

[11]https://github.com/ChrisWaites/adaptive-machine-unlearning

[12]https://www.dropbox.com/s/1lhx962axovbbom/FedEraser-Code.zip?dl=0

[13]https://github.com/fshp971/mcmc-unlearning

preservation of copyright and license notices and includes the permissions of commercial use, modification, distribution, and private use.

To our best knowledge, this work is the first to execute one-time operation of simultaneous training and unlearning in advance for a series of machine unlearning requests, as long as the actual data removals are below the certified budget of data removals, while there is no need to know the forgotten data, by leveraging the theory of randomized smoothing and gradient quantization. Many machine learning applications often need to collect massive amount of data from third parties for model training. This raises a legitimate privacy risk: training data can be practically reconstructed from models [35, 112, 125, 7, 85, 87, 14]. In addition, modern privacy regulations, such as the European Union's General Data Protection Regulation (GDPR) [98] and the California Consumer Privacy Act (CCPA) [71], enforce the right to be forgotten, i.e., entitle data owners to the right to have their private data removed at their requests [87, 83, 20]. Our framework is able to resolve the requests of data removal in a timely and cost-efficient manner. Our framework can play an important building block for a wide variety of privacy-critical applications that usually require near-zero tolerance of data leaking, such as financial and health data analyses. This paper is primarily of a theoretical nature. We expect our findings to produce positive impact, i.e, significantly improve the efficiency of machine unlearning models by simultaneously training and unlearning in advance. To our best knowledge, we do not envision any immediate negative societal impacts of our results, such as security, privacy, and fairness issues.

An important product of this paper is to explore the possibility of simultaneous training and unlearning in advance as well as one-time unlearning. Due to high-dimensional double integrals or non-integrable mapping between samples and labels in the randomized data smoothing and gradient quantization method, the randomized gradient smoothing and quantization approach is designed to produce high confidence certificates for the certified machine unlearning. Our theoretical framework can inspire further improved development and implementations on certified machine unlearning with better applicability and efficiency from the academic institutions and industrial research labs.