# OpenReview forum: "Prompt Certified Machine Unlearning with Randomized Gradient Smoothing and Quantization"
_NeurIPS.cc/2022/Conference — NeurIPS 2022 Accept_

### Official Review · Reviewer_VAAx · 2022-07-10

**Rating:** 8
**Confidence:** 4
**Soundness:** 4 excellent
**Presentation:** 3 good
**Contribution:** 4 excellent

**Summary:**

The authors propose a novel certified machine unlearning algorithm to improve the unlearning efficiency for complex models on large-scale data. First, the authors present an analytic framework to connect  randomized smoothing for certified robustness on classification to randomized smoothing for certified machine unlearning on gradient quantization. Second, the paper develops a prompt certified machine unlearning model for producing the effective certificates of data removals based on randomized data smoothing and gradient quantization. Finally, the authors propose a practical framework of randomized gradient smoothing and quantization for producing the high confidence certificates in an efficient manner.

The proposed PCMU method brings three significant benefits: simultaneously conduct the training and unlearning for improving the unlearning efficiency, one-time training for responding multiple machine unlearning requests at a time, and no need to know the forgotten data before the unlearning.


**Questions:**

See the above weaknesses.


**Strengths And Weaknesses:**

Strengths:

+ The authors study an important research problem, i.e., prompt machine unlearning, which is important to improve the unlearning efficiency for complex models on large-scale data and to provide the timely response to a series of machine unlearning requests. There are few prior works to conduct this problem.

+ The motivation for proposing randomized gradient smoothing and quantization techniques is clearly explained. The method and the claim are correct and sound. The authors provide enough methodology description and theoretical analysis to explain their proposed PCMU model.

+ The paper conducts the theoretical analysis to derive the certified radius regarding the data change, the certified budget of data removals, and the correlation between two types of certified radii in two frameworks. This work integrates the certifying and training of machine unlearning into a uniﬁed framework for further enhancing the unlearning performance. The convergence analysis is conducted to demonstrate the effectiveness and efficiency of the prompt certified machine unlearning algorithm.

+ The paper provides comprehensive extensive evaluation on three real datasets to demonstrate the superior performance of the proposed techniques against a number of SOTA baselines. The experiment results look promising.

Weaknesses:
- There are several typos and the paper would beneﬁt of a careful proofread. For example, "which are used to derive the certified budget B about R'" in P2 -> "B'", "the certified budget of data removal" in P5 -> "removals", and "Notice that the accuracy and error on test data by our PCMU keeps unchanged" in P8 -> "keep".

- It would be nice to move the related work section to the paper for readers to better understand and appreciate the technical contributions of this paper compared with existing studies, instead of the appendix.

- In the experiments, Tables 1-2 and relevant texts lie in different pages, which decreases the paper’s readability. I suggest the authors to update the paper layout to get more clear presentation.

---

> ### Author Response · Authors · 2022-08-02
> **Point-by-point response to the comments made by Reviewer VAAx**
>
> We thank this reviewer for the encouraging comments. We are delighted to hear your positive comments on our contribution to the theoretical explanation of the certified machine unlearning. We theoretically derive the certified radii $R$ and $R'$ regarding the data change before and after data removals and the corresponding certified budgets $B$ and $B'$ of data removals about $R$ and $R'$ in two certified machine unlearning methods respectively. Most importantly, we recognize the correlations between $R$ and $R'$ and between $B$ and $R$, which are used to derive the certified budget $B'$ about $R'$. Thus, our proposed prompt certified machine unlearning method, PCMU, provides a tight guarantee of certified machine unlearning: the parameters and gradients of the learnt machine unlearning model keep unchanged against the data removals within the certified budget.
>
> **Weaknesses 1:** There are several typos and the paper would benefit of a careful proofread. For example, "which are used to derive the certified budget B about R'" in P2 -> "B'", "the certified budget of data removal" in P5 -> "removals", and "Notice that the accuracy and error on test data by our PCMU keeps unchanged" in P8 -> "keep".
>
> **Weaknesses 2:** It would be nice to move the related work section to the paper for readers to better understand and appreciate the technical contributions of this paper compared with existing studies, instead of the appendix.
>
> **Weaknesses 3:** In the experiments, Tables 1-2 and relevant texts lie in different pages, which decreases the paper’s readability. I suggest the authors to update the paper layout to get more clear presentation.
>
> **Answer**: Thanks for your helpful comments. We have proofread our manuscript and corrected the above grammar typos in the rebuttal revision and reorganized the experiment tables and associated texts for improving the paper’s readability. In addition, we will move the related work to the paper.

---

> > ### Comment · Reviewer_VAAx · 2022-08-08
> > **Post-rebuttal comments**
> >
> > I have read the rebuttal and I'm satisfied with the authors' responses as all my concerns have been addressed. I'd like to maintain my score.

---

> > > ### Author Response · Authors · 2022-08-08
> > > **Response to Reviewer VAAx**
> > >
> > > Thanks again for taking the time to review our submission! We will include all the discussions and revisions in this rebuttal into the submission.

---

### Official Review · Reviewer_JFbC · 2022-07-11

**Rating:** 8
**Confidence:** 3
**Soundness:** 4 excellent
**Presentation:** 3 good
**Contribution:** 4 excellent

**Summary:**

This paper presents a novel certified machine unlearning framework that targets the issue in the expensive computational cost of training and unlearning. The authors analogize certified robustness on classification against adversarial attacks to certified machine unlearning on gradient quantization against data removals. The randomized gradient smoothing and quantization techniques are developed to guarantee that the learnt model shares the same gradients (and parameters) and has the same performance with the naive unlearning model retrained on only the remembered data, with only the cost of simultaneous training and unlearning. The theoretical analyses validate the effectiveness of certified machine unlearning in terms of the certified radius and the certified budget of data removals. Overall, the studied problem is interesting and practically important. The experimental results look promising.


**Questions:**

Data removal description and other smoothing strategies.

**Ethics Review Area:**

["I don’t know"]

**Limitations:**

It is unclear of the potential negative societal impacts of the results, such as security, privacy, and fairness issues, etc.

**Strengths And Weaknesses:**

Strengths:

1. Existing machine unlearning methods separate the unlearning process into two sequential operations of training and unlearning, which leads to non-trivial computation cost when training complex models over large datasets. In addition, these methods often sequentially address multiple unlearning requests one by one. To improve the unlearning efficiency, this work trains and unlearns the model simultaneously.

2. The authors propose a randomized gradient smoothing and quantization technique to directly train an unlearning model in advance with fast convergence and certified unlearning guarantees. The framework is able to resolve the requests of data removal in a timely and cost-efficient manner.

3. The proposed method provides a general machine unlearning framework. The proposed framework is important for privacy-critical applications that usually require near-zero tolerance of data leaking, such as financial and health data analyses.

4. This work theoretically analyzes and understands the certified radius regarding the data change before and after data removals and the certified budget of data removals in machine unlearning. Extensive experimental results on different benchmark datasets have been conducted to validate the efficacy of the developed prompt certified machine unlearning algorithms.

Weaknesses:

1. The paper provides descriptions of the benchmark image classification datasets and learning models in the paper and appendix. It would be nice to include the description of the data removals for how to separate the datasets into the forgotten data and the remembered data for each benchmark dataset.

2. It would be interesting to see the results about different smoothing strategies exploited in certified machine unlearning problems, such as Laplacian and uniform smoothing.

---

> ### Author Response · Authors · 2022-08-02
> **Point-by-point response to the comments made by Reviewer JFbC**
>
> We thank this reviewer for the constructive comments. We are delighted to hear your positive comments on our contribution to the prompt machine unlearning with fast convergence and certified unlearning guarantees. Following your precise assessment, we believe that our work makes a solid step to simultaneously execute the training and unlearning operations for improving the unlearning efficiency for complex models on large-scale data. We also want to remark that the simultaneous training and unlearning is a one-time operation that can provide the timely response to a series of machine unlearning requests, as long as the actual data removals are below the certified budget of data removals.
>
> **Weaknesses 1:** The paper provides descriptions of the benchmark image classification datasets and learning models in the paper and appendix. It would be nice to include the description of the data removals for how to separate the datasets into the forgotten data and the remembered data for each benchmark dataset.
>
> **Answer:** Thanks for the kind suggestion. In this work, by following several representative machine unlearning methods [1, 2, 3], where each learning request is modeled as a random draw from the training data in terms of a uniform distribution. Given a ratio of data removal, the forgotten data $D_f$ are sampled uniformly from the complete training data $D$ with this ratio. The remaining dataset $D_r$ (i.e., $D = D_f \cup D_r$) will be considered as the remembered data. This sampling approach is more realistic since a removal request may be applied to any data examples with the same probability.
>
> [1] Lucas Bourtoule, Varun Chandrasekaran, Christopher A. Choquette-Choo, Hengrui Jia, Adelin Travers, Baiwu Zhang, David Lie, and Nicolas Papernot. Machine unlearning. In S\&P 2021.
>
> [2] Varun Gupta, Christopher Jung, Seth Neel, Aaron Roth, Saeed Sharifi-Malvajerdi, and Chris Waites. Adaptive machine unlearning. In NeurIPS 2021.
>
> [3] Ga Wu, Masoud Hashemi, and Christopher Srinivasa. PUMA: performance unchanged model augmentation for training data removal. In AAAI 2022.
>
> **Weaknesses 2:** It would be interesting to see the results about different smoothing strategies exploited in certified machine unlearning problems, such as Laplacian and uniform smoothing.
>
> **Answer:** Thanks for the thoughtful comment. The following tables evaluate the impact of different smoothing techniques on our proposed prompt certified machine unlearning method, PCMU, over two popular image classification datasets: Fashion-MNIST and CIFAR-10. We replace the randomized smoothing component in the original PCMU model with Laplacian smoothing and uniform smoothing respectively. It is observed that two PCMU variants with Laplacian smoothing and uniform smoothing achieve the close performance to the original PCMU model with randomized smoothing, showing the generality of PCMU to the machine unlearning. Compared with the results achieved by the nine state-of-the-art baselines in Tables 2 and 4 in the submission, two PCMU variants still substantially outperform the performance of other baselines in most experiments. We have included all the experiment results in this rebuttal into the rebuttal revision.
>
> **Performance with 20\% data removal and CNN on Fashion-MNIST**
>
> Metric | $Accuracy$ | $Error_t$ | $Error_r$ | $Error_f$ | Training Runtime (s) | Unlearning Runtime (s) | Total Runtime (s)
> ---|---|---|---|---|---|---|---
> Retrain | 88.21 | 11.79 | 9.75 | 11.76 | 687 | 561 | 1,248
> PCMU | **88.34** | **11.66** | **10.25** | **11.47** | 802 | 0 | 802 |
> PCMU+Laplacian Smoothing | 86.16 | 13.84 | 12.29 | 13.20 | 773 | 0 | 773 |
> PCMU+Uniform Smoothing | 86.86 | 13.14 | 10.90 | 12.77 | 804 | 0 | 804 |
>
> **Performance with 20\% data removal and LeNet on CIFAR-10**
> Metric | $Accuracy$ | $Error_t$ | $Error_r$ | $Error_f$ | Training Runtime (s) | Unlearning Runtime (s) | Total Runtime (s)
> ---|---|---|---|---|---|---|---
> Retrain | 63.29 | 36.71 | 24.59 | 36.89 | 846 | 673 | 1,519
> PCMU | **64.33** | **35.67** | **25.18** | **35.32** | 903 | 0 | 903 |
> PCMU+Laplacian Smoothing | 61.77 | 38.23 | 29.41 | 38.30 | 928 | 0 | 928 |
> PCMU+Uniform Smoothing | 62.17 | 37.83 | 28.06 | 38.60 | 936 | 0 | 936 |

---

> > ### Comment · Reviewer_JFbC · 2022-08-09
> > **Response**
> >
> > Thanks for the authors' response, which cleared all of my questions. It is a cool work to enable removed data-agnostic simultaneous training and unlearning for better unlearning efficiency and generality.

---

### Official Review · Reviewer_xTHV · 2022-07-11

**Rating:** 3
**Confidence:** 3
**Soundness:** 2 fair
**Presentation:** 2 fair
**Contribution:** 2 fair

**Summary:**

This papers tackles the problem of machine unlearning, that is, removing training data upon request from a model, as if the model is trained only with the retained data. The proposed approach, PCMU, trains a model with randomly smoothed quantized gradient. Analogous to certified robustness, the paper presents some certified radius of the proposed method, against to pertubation of the gradient. Unlike previous machine unlearning approaches, PCMU directly trains a model that is robust to data removal, so it does not require an additional unlearning stage. Experimental results show that PCMU can achieve a more similar error rate with retraining than competitive models.

**Questions:**

1. Does the proposed approach work for non-uniform data removal? For example, a "group" or class of samples are removed completely.
2. In Eq. (21), I suppose \bar G is a gradient, and \epsilon ~ \mathcal D is a datum. What does it mean by adding a data and a gradient?
3. What does "prompt" in the title mean?

Typos:
L33: "certified" should be capatalized.
L53: two consecutive periods.

**Limitations:**

There are not apparent further limitations or potential  negative societal impacts other than those I raised for weaknesses and questions.

**Strengths And Weaknesses:**

Strength
- some theoretical results are provided
- the proposed method is faster than previous methods, and it can matches the accuracy better with the retraining approach

Weaknesses
- I'm not an expert on machine unlearning. But from my perspective, rather than a machine unlearning method, the proposed approach looks more like a differential privacy work, which trains a model without utilizing too specific features from individual users, so the trained model automatically satisfies removal requirements. I wonder whether it counts as machine unlearning.
- The writing is somewhat vague: there is not a description or pseudocode of how the training and unlearning are performed, making the paper difficult to follow.

---

> ### Author Response · Authors · 2022-08-02
> **Point-by-point response to the comments made by Reviewer xTHV**
>
> We thank this reviewer for the great suggestion!
>
> **Weaknesses 1:** I'm not an expert on machine unlearning. But from my perspective, rather than a machine unlearning method, the proposed approach looks more like a differential privacy work, which trains a model without utilizing too specific features from individual users, so the trained model automatically satisfies removal requirements. I wonder whether it counts as machine unlearning.
>
> **Answer:** Thanks for the thoughtful comment.
> Differential privacy [1,2] and randomized smoothing [3] are two orthogonal techniques. To our best knowledge, our PCMU method is the first to propose randomized gradient smoothing and quantization techniques for certified machine unlearning. One main strength of our PCMU method is to **derive the certified radius and the certified budget of data removals for the certified machine unlearning**. Another unique power is to **simultaneously execute the training and unlearning operations for dramatically improving the unlearning efficiency**.
>
> In this work, we establish a connection between randomized smoothing for certified robustness on classification and randomized smoothing for certified machine unlearning on gradient quantization. We analogize the data removals on the entire training data (i.e., the perturbations on the entire data) in the machine unlearning to the adversarial attacks (i.e., the perturbations on the data samples) in the certified robustness and liken the output quantized gradients in the former to the output discrete class labels in the latter. Since the output class labels in the latter through randomized smoothing are able to keep unchanged and correct against adversarial attacks within the certified radius, it is highly possible that the output quantized gradients in the former through randomized smoothing can keep unchanged against data removals within the certified budget, which implies that the learnt model shares the same gradients (and parameters) with the naive one retrained on only the remembered data.
>
> [1] Cynthia Dwork. Differential Privacy. ICALP (2) 2006: 1-12.
>
> [2] Cynthia Dwork: Differential Privacy: A Survey of Results. TAMC 2008: 1-19.
>
> [3] J. M. Cohen, E. Rosenfeld, and J. Z. Kolter. Certified adversarial robustness via randomized smoothing. In ICML 2019.
>
> **Weaknesses 2:** The writing is somewhat vague: there is not a description or pseudocode of how the training and unlearning are performed, making the paper difficult to follow.
>
> **Answer:** Thanks for the valuable suggestion. The following are the detailed descriptions of our PCMU method step by step: (1) Train the model in a usual manner with loss function $\mathcal{L}$, e.g., cross-entropy for image classification, and model parameter $w$; (2) Calculate the gradient $G(x,y) = \frac{\partial \mathcal{L}(x, y; w)} {\partial w}$ in Eq.(5) in the submission; (3) Compute  the gradient average $\bar G$ in terms of the gradient $G(x_i,y_i)$ of each sample $(x_i,y_i)$ in Eq.(20) in the submission; (4) quantize each dimension $t$ $(t=1,\cdots,T)$ of the continuous gradient plus Gaussian noise $Q^t(\bar G+\varepsilon)$ in Eq.(21) over a discrete three-class space $\{-1, 0, 1\}$, for mimicking the classification in the randomized smoothing for certified robustness based on Eq.(6) in the submission; (5) Perform the randomized gradient smoothing for certified machine unlearning $S^{t\prime}(\bar G) = \underset{c \in \{-1, 0, 1\}} {\arg\max} \underset{\varepsilon \sim \mathcal{D}} {\mathbb{P}} (Q^t(\bar G+\varepsilon)=c)$ in Eq.(21) in the submission; (6) Derive the certified radius $R^{\prime}$ in Eq.(24) and the certified budget $B^{\prime}$ of data removals in Eq.(26) in the submission; (7) Integrate the model training for a specific learning task (e.g., image classification), randomized gradient smoothing, and gradient quantization into a unified framework for directly training a machine unlearning model with the data removal certificates as a guidance, for guaranteeing that the model parameters and gradients keep unchanged against the data removals within the certified budget, in terms of $w = w - \eta [S^{1\prime}(\bar G), \cdots, S^{T\prime}(\bar G)]$ with smoothed and quantized gradients in Eq.(29) in the submission; and (8) Enter the next training round until convergence.

---

> > ### Author Response · Authors · 2022-08-02
> > **Point-by-point response to the comments made by Reviewer xTHV (Continued)**
> >
> > **Question 1:** Does the proposed approach work for non-uniform data removal? For example, a "group" or class of samples are removed completely.
> >
> > **Answer:** Thanks for the kind suggestion. In the submission, all the formulae, methods, and theories do not have assumptions or restrictions regarding the distribution of data removals. Thus, our proposed prompt certified machine unlearning method, PCMU, is able to work on the case of removals of class samples, as long as the actual data removals are below the certified budget of data removals. In fact, the removals of arbitrary data samples in our experiments is more general than the removals of class samples.
> >
> > In order to validate the performance of our PCMU for this special machine unlearning problem, we randomly choose one class and remove all samples from this class over CIFAR-10. The following table exhibits the corresponding experiment results, which demonstrate that PCMU is also effective as well as efficient for addressing this special machine unlearning problem. We have included all the experiment results in this rebuttal into the rebuttal revision.
> >
> > **Performance with data removal of samples from one class and LeNet on CIFAR-10**
> > Metric | $Accuracy$ | $Error_t$ | $Error_r$ | $Error_f$ | Training Runtime (s) | Unlearning Runtime (s) | Total Runtime (s)
> > ---|---|---|---|---|---|---|---
> > Retrain | 58.91 | 41.09 | 28.23 | 100 | 859 | 738 | 1,597 |
> > Fisher | 62.28 | 37.72 | 34.28 | 29.51 | 1,459 | 1,490 | 2,949 |
> > certified removal | 38.40 | 61.61 | 61.73 | 60.32 | **886** | 218 | 1,104 |
> > NTK | 62.71 | 37.29 | 32.47 | 42.20 | 1,459 | 1,353 | 2,812 |
> > MCMC unlearning | 57.13 | 42.87 | 16.44 | 49.52 | 1,565 | 734 | 2,299 |
> > PCMU | **60.15** | **39.85** | **26.51** | **72.18** | 926 | **0** | **926** |
> >
> > **Question 2:** In Eq.(21), I suppose $\bar G$ is a gradient, and $\epsilon \sim \mathcal D$ is a datum. What does it mean by adding a data and a gradient?
> >
> > **Answer:** Thanks for the insightful question. Sorry for the confusion caused by our notation of two similar symbols. We use $D$ to denote the complete training dataset, which is defined in Line 119 in the submission. We use $\mathcal{D} = \mathcal{N}\left(0, \sigma^{2} I\right)$ to denote a Gaussian distribution, whose definition is in Line 164 in the submission. Thus, in Eq.(21), $\epsilon \sim \mathcal D$ represents the Gaussian noise to be added to the gradient $\bar G$ for randomized gradient smoothing.
> >
> > **Question 3:** What does "prompt" in the title mean?
> >
> > **Answer:** Thanks for the comment. To our best knowledge, most of existing machine unlearning methods consist of two sequential operations: (1) Training: train a machine learning model on the complete training data and (2) Unlearning: generate an unlearning model from the former. The combination of two operations is computationally expensive when training complex models over large datasets. In addition, they often sequentially redo the unlearning operations one by one, when addressing a series of machine unlearning requests.
> >
> > Our PCMU method incorporates the sequential training and unlearning processes into a unified model to directly learn an unlearning model, i.e., **simultaneously execute the training and unlearning operations**, which is able to dramatically improve the unlearning efficiency for complex models on large-scale data. In addition, our PCMU method performs the **one-time operation of simultaneous training and unlearning** based on the derived certified budget of data removals in the submission. Thus, our PCMU method can provide the timely response to a series of machine unlearning requests, as long as the actual data removals are below the certified budget of data removals. Namely, our PCMU method needs time to train the model but does not have additional time for machine unlearning. The superior efficiency of PCMU has been reflected in Tables 1-15 in the submission. We hope these new technical clarifications and experiment results have addressed your concerns about our proposed prompt certified machine unlearning method, PCMU.

---

> > > ### Author Response · Authors · 2022-08-08
> > > **Thanks to Reviewer xTHV**
> > >
> > > We would like to thank the reviewer for taking the time to review our paper and for the valuable comments. We hope our response has adequately addressed your concerns raised in the review.
> > >
> > > We truly appreciate this opportunity to improve our work and shall be most grateful for any feedback you could give to us.
> > >
> > > The following is the summary of our paper's contributions:
> > >
> > > **(1) Simultaneous training and unlearning for better unlearning efficiency.**
> > >
> > > **(2) Agnostic to the removed/forgotten data before performing the unlearning operation.**
> > >
> > > **(3) Derivation of the certified radius and the certified budget of data removals for the certified unlearning.**

---

> > > > ### Author Response · Authors · 2022-08-09
> > > > **Post-rebuttal Comments**
> > > >
> > > > Thanks again for taking the time to review our submission!
> > > >
> > > > We believe that our work makes a fundamental contribution to the certified machine unlearning in empirical and theoretical perspective. We have now clarified the significance and the contributions of our PCMU method for prompt certified machine unlearning with more detailed information in our rebuttal summary.
> > > >
> > > > Please kindly let us know if anything is unclear or any constructive post-rebuttal feedbacks are raised.

---

### Author Response · Authors · 2022-08-02
**Common Comments to all Reviewers**

We would like to thank the three reviewers for the helpful and constructive comments. We have tried our best to clarify the concerns and comments by all three reviewers. We have presented the point-by-point response to the comments made by each of the three reviewers. We have included the discussions, analyses, explanations, and experiment results in this rebuttal into the rebuttal revision.

---

> ### Author Response · Authors · 2022-08-05
> **Follow-up response**
>
> Dear NeurIPS 2022 Reviewers, Area Chairs, Senior Area Chairs, and Conference Program Chairs,
>
> Thank you very much for your insightful comments. We have tried our best to clarify and address the concerns and comments raised by the reviewer in the initial response. We are glad to answer and clarify any further questions and advices from the reviewer for better readability.

---

### Meta-Review · Area_Chair_x7xC · 2022-08-27

**Recommendation:** Accept
**Confidence:** Certain

**Metareview:**

This paper proposes an algorithm for simultaneous learning and unlearning without the knowledge of the datapoints that will be forgotten. This reduces the computational cost associated with unlearning in a unified fashion. Both experimental and theoretical results are interesting and the paper would be a great addition to NeurIPS.

**Award:**

No

---

### Decision · Program_Chairs · 2022-09-14

Accept